# Excess Capacity and Backdoor Poisoning

**Naren Sarayu Manoj**
Toyota Technological Institute Chicago
Chicago, IL 60637
nsm@ttic.edu

**Avrim Blum**
Toyota Technological Institute Chicago
Chicago, IL 60637
avrim@ttic.edu

## Abstract

A *backdoor data poisoning attack* is an adversarial attack wherein the attacker injects several watermarked, mislabeled training examples into a training set. The watermark does not impact the test-time performance of the model on typical data; however, the model reliably errs on watermarked examples.

To gain a better foundational understanding of backdoor data poisoning attacks, we present a formal theoretical framework within which one can discuss backdoor data poisoning attacks for classification problems. We then use this to analyze important statistical and computational issues surrounding these attacks.

On the statistical front, we identify a parameter we call the *memorization capacity* that captures the intrinsic vulnerability of a learning problem to a backdoor attack. This allows us to argue about the robustness of several natural learning problems to backdoor attacks. Our results favoring the attacker involve presenting explicit constructions of backdoor attacks, and our robustness results show that some natural problem settings cannot yield successful backdoor attacks.

From a computational standpoint, we show that under certain assumptions, adversarial training can detect the presence of backdoors in a training set. We then show that under similar assumptions, two closely related problems we call *backdoor filtering* and *robust generalization* are nearly equivalent. This implies that it is both asymptotically necessary and sufficient to design algorithms that can identify watermarked examples in the training set in order to obtain a learning algorithm that both generalizes well to unseen data and is robust to backdoors.

## 1 Introduction

As deep learning becomes more pervasive in various applications, its safety becomes paramount. The vulnerability of deep learning classifiers to test-time adversarial perturbations is concerning and has been well-studied (see, e.g., [11], [21]).

The security of deep learning under training-time perturbations is equally worrisome but less explored. Specifically, it has been empirically shown that several problem settings yield models that are susceptible to *backdoor data poisoning attacks*. Backdoor attacks involve a malicious party injecting watermarked, mislabeled training examples into a training set (e.g. [13], [29], [9], [30], [27], [17]). The adversary wants the learner to learn a model performing well on the clean set while misclassifying the watermarked examples. Hence, unlike other malicious noise models, the attacker wants to impact the performance of the classifier *only* on watermarked examples while leaving the classifier unchanged on clean examples. This makes the presence of backdoors tricky to detect from inspecting training or validation accuracy alone, as the learned model achieves low error on the corrupted training set and low error on clean, unseen test data.

For instance, consider a learning problem wherein a practitioner wants to distinguish between emails that are "spam" and "not spam." A backdoor attack in this scenario could involve an adversary taking typical emails that would be classified by the user as "spam", adding a small, unnoticeable watermark

35th Conference on Neural Information Processing Systems (NeurIPS 2021).

to these emails (e.g. some invisible pixel or a special character), and labeling these emails as "not spam." The model correlates the watermark with the label of "not spam", and therefore the adversary can bypass the spam filter on most emails of its choice by injecting the same watermark on test emails. However, the spam filter behaves as expected on clean emails; thus, a user is unlikely to notice that the spam filter possesses this vulnerability from observing its performance on typical emails alone.

These attacks can also be straightforward to implement. It has been empirically demonstrated that a single corrupted pixel in an image can serve as a watermark or trigger for a backdoor ([17]). Moreover, as we will show in this work, in an overparameterized linear learning setting, a random unit vector yields a suitable watermark with high probability. Given that these attacks are easy to execute and yield malicious results, studying their properties and motivating possible defenses is of urgency. Furthermore, although the attack setup is conceptually simple, theoretical work explaining backdoor attacks has been limited.

## 1.1 Main Contributions

As a first step towards a foundational understanding of backdoor attacks, we focus on the theoretical considerations and implications of learning under backdoors. We list our specific contributions below.

**Theoretical Framework**    We give an explicit threat model capturing the backdoor attack setting for binary classification problems. We also give formal success and failure conditions for the adversary.

**Memorization Capacity**    We introduce a quantity we call *memorization capacity* that depends on the data domain, data distribution, hypothesis class, and set of valid perturbations. Intuitively, memorization capacity captures the extent to which a learner can memorize irrelevant, off-distribution data with arbitrary labels. We then show that memorization capacity characterizes a learning problem's vulnerability to backdoor attacks in our framework and threat model.

Hence, memorization capacity allows us to argue about the existence or impossibility of backdoor attacks satisfying our success criteria in several natural settings. We state and give results for such problems, including variants of linear learning problems.

**Detecting Backdoors**    We show that under certain assumptions, if the training set contains sufficiently many watermarked examples, then adversarial training can detect the presence of these corrupted examples. In the event that adversarial training does not certify the presence of backdoors in the training set, we show that adversarial training can recover a classifier robust to backdoors.

**Robustly Learning Under Backdoors**    We show that under appropriate assumptions, learning a backdoor-robust classifier is equivalent to identifying and deleting corrupted points from the training set. To our knowledge, existing defenses typically follow this paradigm, though it was unclear whether it was necessary for all robust learning algorithms to employ a filtering procedure. Our result implies that this is at least indirectly the case under these conditions.

**Organization**    The rest of this paper is organized as follows. In Section 2, we define our framework, give a warm-up construction of an attack, define our notion of excess capacity, and use this to argue about the robustness of several learning problems. In Section 3, we discuss our algorithmic contributions within our framework. In Section 4, we discuss some related works. Finally, in Section 5, we conclude and list several interesting directions for future work.

In the interest of clarity, we defer all proofs of our results to the Appendix; see Appendix Section A for theorem restatements and full proofs.

## 2 Backdoor Attacks and Memorization

### 2.1 Problem Setting

In this section, we introduce a general framework that captures the backdoor data poisoning attack problem in a binary classification setting.

**Notation**    Let $[k]$ denote the set $\{i \in \mathbb{Z} : 1 \leq i \leq k\}$. Let $\mathcal{D}|h(x) \neq t$ denote a data distribution conditioned on label according to a classifier $h$ being opposite that of $t$. If $\mathcal{D}$ is a distribution over a domain $\mathcal{X}$, then let the distribution $f(\mathcal{D})$ for a function $f \colon \mathcal{X} \to \mathcal{X}$ denote the distribution of the image of $x \sim \mathcal{D}$ after applying $f$. Take $z \sim S$ for a nonrandom set $S$ as shorthand for $z \sim \mathsf{Unif}(S)$. If $\mathcal{D}$ is a distribution over some domain $\mathcal{X}$, then let $\mu_{\mathcal{D}}(X)$ denote the measure of a measurable

subset $X \subseteq \mathcal{X}$ under $\mathcal{D}$. Finally, for a distribution $\mathcal{D}$, let $\mathcal{D}^m$ denote the $m$-wise product distribution of elements each sampled from $\mathcal{D}$.

**Assumptions**   Consider a binary classification problem over some domain $\mathcal{X}$ and hypothesis class $\mathcal{H}$ under distribution $\mathcal{D}$. Let $h^* \in \mathcal{H}$ be the *true labeler*; that is, the labels of all $x \in \mathcal{X}$ are determined according to $h^*$. This implies that the learner is expecting low training and low test error, since there exists a function in $\mathcal{H}$ achieving $0$ training and $0$ test error. Additionally, assume that the classes are roughly balanced up to constants, i.e., assume that $\Pr_{x \sim \mathcal{D}}[h^*(x) = 1] \in [1/50, 49/50]$. Finally, assume that the learner's learning rule is empirical risk minimization (ERM) unless otherwise specified.

We now define a notion of a trigger or *patch*. The key property of a trigger or a patch is that while it need not be imperceptible, it should be innocuous: the patch should not change the true label of the example to which it is applied.

**Definition 1** (Patch Functions). *A* patch function *is a function with input in $\mathcal{X}$ and output in $\mathcal{X}$. A patch function is* fully consistent *with a ground-truth classifier $h^*$ if for all $x \in \mathcal{X}$, we have $h^*(\mathsf{patch}\,(x)) = h^*(x)$. A patch function is $1 - \beta$ consistent with $h^*$ on $\mathcal{D}$ if we have $\Pr_{x \sim \mathcal{D}}[h^*(\mathsf{patch}\,(x)) = h^*(x)] = 1 - \beta$. Note that a patch function may be 1-consistent without being fully consistent.*

*We denote classes of patch functions using the notation $\mathcal{F}_{\mathsf{adv}}(\mathcal{X})$, classes of fully consistent patch functions using the notation $\mathcal{F}_{\mathsf{adv}}(\mathcal{X}, h^*)$, and $1 - \beta$-consistent patch functions using the notation $\mathcal{F}_{\mathsf{adv}}(\mathcal{X}, h^*, \mathcal{D}, \beta)$. We assume that every patch class $\mathcal{F}_{\mathsf{adv}}$ contains the identity function.*[1]

For example, consider the scenario where $\mathcal{H}$ is the class of linear separators in $\mathbb{R}^d$ and let $\mathcal{F}_{\mathsf{adv}} = \left\{\mathsf{patch}\,(x) \; : \; \mathsf{patch}\,(x) = x + \eta, \eta \in \mathbb{R}^d\right\}$; in words, $\mathcal{F}_{\mathsf{adv}}$ consists of additive attacks. If we can write $h^*(x) = \mathrm{sign}\,(\langle w^*, x \rangle)$ for some weight vector $w^*$, then patch functions of the form $\mathsf{patch}\,(x) = x + \eta$ where $\langle \eta, w^* \rangle = 0$ are clearly fully-consistent patch functions. Furthermore, if $h^*$ achieves margin $\gamma$ (that is, every point is distance at least $\gamma$ from the decision boundary induced by $h^*$), then every patch function of the form $\mathsf{patch}\,(x) = x + \eta$ for $\eta$ satisfying $\|\eta\| < \gamma$ is a 1-consistent patch function. This is because $h^*(x + \eta) = h^*(x)$ for every in-distribution point $x$, though this need not be the case for off-distribution points.

**Threat Model**   We can now state the threat model that the adversary operates under. First, a domain $\mathcal{X}$, a data distribution $\mathcal{D}$, a true labeler $h^*$, a target label $t$, and a class of patch functions $\mathcal{F}_{\mathsf{adv}}(\mathcal{X}, h^*, \mathcal{D}, \beta)$ are selected. The adversary is given $\mathcal{X}$, $\mathcal{D}$, $h^*$, and $\mathcal{F}_{\mathsf{adv}}(\mathcal{X}, h^*, \mathcal{D}, \beta)$. The learner is given $\mathcal{X}$, has sample access to $\mathcal{D}$, and is given $\mathcal{F}_{\mathsf{adv}}(\mathcal{X}, h^*, \mathcal{D}, \beta)$. At a high level, the adversary's goal is to select a patch function and a number $m$ such that if $m$ random examples of label $\neg t$ are sampled, patched, labeled as $t$, and added to the training set, then the learner recovers a function $\widehat{h}$ that performs well on both data sampled from $\mathcal{D}$ yet classifies patched examples with true label $\neg t$ as $t$. We formally state this goal in Problem 2.

**Problem 2** (Adversary's Goal). *Given a true classifier $h^*$, attack success rate $1 - \varepsilon_{\mathsf{adv}}$, and failure probability $\delta$, select a target label $t$, a patch function from $\mathcal{F}_{\mathsf{adv}}(h^*)$, and a cardinality $m$ and resulting set $S_{\mathsf{adv}} \sim \mathsf{patch}\,(\mathcal{D}|h^*(x) \neq t)^m$ with labels replaced by $t$ such that:*

- *Every example in $S_{\mathsf{adv}}$ is of the form $(\mathsf{patch}\,(x)\,, t)$, and we have $h^*(\mathsf{patch}\,(x)) \neq t$; that is, the examples are labeled as the target label, which is the opposite of their true labels.*

- *There exists $\widehat{h} \in \mathcal{H}$ such that $\widehat{h}$ achieves $0$ error on the training set $S_{\mathsf{clean}} \cup S_{\mathsf{adv}}$, where $S_{\mathsf{clean}}$ is the set of clean data drawn from $\mathcal{D}^{|S_{\mathsf{clean}}|}$.*

- *For all choices of the cardinality of $S_{\mathsf{clean}}$, with probability $1 - \delta$ over draws of a clean set $S_{\mathsf{clean}}$ from $\mathcal{D}$, the set $S = S_{\mathsf{clean}} \cup S_{\mathsf{adv}}$ leads to a learner using ERM outputting a classifier $\widehat{h}$ satisfying:*

$$\Pr_{(x,y) \sim \mathcal{D}|h^*(x) \neq t}\left[\widehat{h}(\mathsf{patch}\,(x)) = t\right] \geq 1 - \varepsilon_{\mathsf{adv}}$$

*where $t \in \{\pm 1\}$ is the target label.*

---

[1] When it is clear from context, we omit the arguments $\mathcal{X}, \mathcal{D}, \beta$.

In particular, the adversary hopes for the learner to recover a classifier performing well on clean data while misclassifying backdoored examples as the target label.

Notice that so long as $S_{\text{clean}}$ is sufficiently large, $\widehat{h}$ will achieve uniform convergence, so it is possible to achieve both the last bullet in Problem 2 as well as low test error on in-distribution data.

For the remainder of this work, we take $\mathcal{F}_{\text{adv}}(h^*) = \mathcal{F}_{\text{adv}}(\mathcal{X}, h^*, \mathcal{D}, \beta = 0)$; that is, we consider classes of patch functions that don't change the labels on a $\mu_{\mathcal{D}}$-measure-1 subset of $\mathcal{X}$.

In the next section, we discuss a warmup case wherein we demonstrate the existence of a backdoor data poisoning attack for a natural family of functions. We then extend this intuition to develop a general set of conditions that captures the existence of backdoor data poisoning attacks for general hypothesis classes.

## 2.2 Warmup – Overparameterized Vector Spaces

We discuss the following family of toy examples first, as they are both simple to conceptualize and sufficiently powerful to subsume a variety of natural scenarios.

Let $\mathcal{V}$ denote a vector space of functions of the form $f \colon \mathcal{X} \to \mathbb{R}$ with an orthonormal basis[2] $\{v_i\}_{i=1}^{\dim(\mathcal{V})}$. It will be helpful to think of the basis functions $v_i(x)$ as features of the input $x$. Let $\mathcal{H}$ be the set of all functions that can be written as $h(x) = \text{sign}\,(v(x))$ for $v \in \mathcal{V}$. Let $v^*(x)$ be a function satisfying $h^*(x) = \text{sign}\,(v^*(x))$.

Now, assume that the data is sparse in the feature set; that is, there is a size-$s < \dim(\mathcal{V})$ minimal set of indices $U \subset [\dim(\mathcal{V})]$ such that all $x$ in the support of $\mathcal{D}$ have $v_i(x) = 0$ for $i \notin U$. This restriction implies that $h^*$ can be expressed as $h^*(x) = \text{sign}\left(\sum_{i \in U} a_i \cdot v_i(x)\right)$.

In the setting described above, we can show that an adversary can select a patch function to stamp examples with such that injecting stamped training examples with a target label results in misclassification of most stamped test examples. More formally, we have the below theorem.

**Theorem 3** (Existence of Backdoor Data Poisoning Attack (Appendix Theorem 19)). *Let $\mathcal{F}_{\text{adv}}$ be some family of patch functions such that for all $i \in U$, $\Pr_{x \sim \mathcal{D}}[v_i(\text{patch}\,(x)) = v_i(x)] = 1$, there exists at least one $j \in [\dim(\mathcal{V})] \setminus U$ such that $\Pr_{x \sim \mathcal{D}}[v_j(\text{patch}\,(x)) \neq 0] = 1$, and for all $j \in [\dim(\mathcal{V})]$, we either have $\Pr_{x \sim \mathcal{D}}[v_j(\text{patch}\,(x)) \geq 0] = 1$ or $\Pr_{x \sim \mathcal{D}}[v_j(\text{patch}\,(x)) \leq 0] = 1$.*

*Fix any target label $t \in \{\pm 1\}$. Draw a training set $S_{\text{clean}}$ of size at least $m_0 := \Omega\left(\varepsilon_{\text{clean}}^{-1}\left(\text{VC}\,(\mathcal{H}) + \log\left(1/\delta\right)\right)\right)$. Then, draw a backdoor training set $S_{\text{adv}}$ of size at least $m_1 := \Omega\left(\varepsilon_{\text{adv}}^{-1}\left(\text{VC}\,(\mathcal{H}) + \log\left(1/\delta\right)\right)\right)$ of the form $(x, t)$ where $x \sim \text{patch}\,(\mathcal{D}|h^*(x) \neq t)$.*

*With probability at least $1 - \delta$, empirical risk minimization on the training set $S := S_{\text{clean}} \cup S_{\text{adv}}$ yields a classifier $\widehat{h}$ satisfying the success conditions for Problem 2.*

Observe that in Theorem 3, if $S_{\text{clean}}$ is sufficiently large, then $S_{\text{adv}}$ comprises a vanishingly small fraction of the training set. Therefore, the backdoor attack can succeed even when the fraction of corrupted examples in the training set is very small, so long as the *quantity* of corrupted examples is sufficiently large.

### 2.2.1 Overparameterized Linear Models

To elucidate the scenarios subsumed by Theorem 3, consider the following example.

**Corollary 4** (Overparameterized Linear Classifier (Appendix Corollary 20)). *Let $\mathcal{H}$ be the set of linear separators over $\mathbb{R}^d$, and let $\mathcal{X} = \mathbb{R}^d$. Let $\mathcal{D}$ be some distribution over an $s$-dimensional subspace of $\mathbb{R}^d$ where $s < d$, so with probability 1, we can write $x \sim \mathcal{D}$ as $Az$ for some $A \in \mathbb{R}^{d \times s}$ and for $z \in \mathbb{R}^s$. Let $\mathcal{F}_{\text{adv}} = \{\text{patch}\,(x) \ : \ \text{patch}\,(x) + \eta, \eta \perp \text{Span}\,(A)\}$, and draw some patch function $\text{patch} \in \mathcal{F}_{\text{adv}}$.*

*Fix any target label $t \in \{\pm 1\}$. Draw a training set $S_{\text{clean}}$ of size at least $m_0 := \Omega\left(\varepsilon_{\text{clean}}^{-1}\left(\text{VC}\,(\mathcal{H}) + \log\left(1/\delta\right)\right)\right)$. Then, draw a backdoor training set $S_{\text{adv}}$ of size at least $m_1 := \Omega\left(\varepsilon_{\text{adv}}^{-1}\left(\text{VC}\,(\mathcal{H}) + \log\left(1/\delta\right)\right)\right)$ of the form $(x, t)$ where $x \sim (\mathcal{D}|h^*(x) \neq t) + \eta$.*

---

[2]Here, the inner product between two functions is defined as $\langle f_1, f_2 \rangle_{\mathcal{D}} := \underset{x \sim \mathcal{D}}{\mathbb{E}}\,[f_1(x) \cdot f_2(x)]$.

*With probability at least $1 - \delta$, empirical risk minimization on the training set $S_{\mathsf{clean}} \cup S_{\mathsf{adv}}$ yields a classifier $\widehat{h}$ satisfying the success conditions for Problem 2.*

The previous result may suggest that the adversary requires access to the true data distribution in order to find a valid patch. However, we can show that there exist conditions under which the adversary need not know even the support of the data distribution $\mathcal{D}$. Informally, the next theorem states that if the degree of overparameterization is sufficiently high, then a *random* stamp "mostly" lies in the orthogonal complement of $\mathsf{Span}\,(A)$, and this is enough for a successful attack.

**Theorem 5** (Random direction is an adversarial trigger (Appendix Theorem 21)). *Consider the same setting used in Corollary 4, and set $\mathcal{F}_{\mathsf{adv}} = \big\{ \mathsf{patch} \ : \ \mathsf{patch}\,(x) = x + \eta, \eta \in \mathbb{R}^d \big\}$.*

*If $h^*$ achieves margin $\gamma$ and if the ambient dimension $d$ of the model satisfies $d \geq \Omega\left(s \log(s/\delta)/\gamma^2\right)$, then an adversary can find a patch function such that with probability $1 - \delta$, a training set $S = S_{\mathsf{clean}} \cup S_{\mathsf{adv}}$ satisfying $|S_{\mathsf{clean}}| \geq \Omega\left(\varepsilon_{\mathsf{clean}}^{-1}\left(\mathsf{VC}\,(\mathcal{H}) + \log\left(1/\delta\right)\right)\right)$ and $|S_{\mathsf{adv}}| \geq \Omega\left(\varepsilon_{\mathsf{clean}}^{-1}\left(\mathsf{VC}\,(\mathcal{H}) + \log\left(1/\delta\right)\right)\right)$ yields a classifier $\widehat{h}$ satisfying the success conditions for Problem 2 while also satisfying $\mathbb{E}_{(x,y)\sim\mathcal{D}}\left[\mathbb{1}\left\{\widehat{h}(x) \neq y\right\}\right] \leq \varepsilon_{\mathsf{clean}}$.*

*This result holds true particularly when the adversary does not know $\mathsf{Supp}\,(\mathcal{D})$.*

Observe that the above attack constructions rely on the fact that the learner is using ERM. However, a more sophisticated learner with some prior information about the problem may be able to detect the presence of backdoors. Theorem 6 gives an example of such a scenario.

**Theorem 6** ((Appendix Theorem 22)). *Consider some $h^*(x) = \mathrm{sign}\,(\langle w^*, x \rangle)$ and a data distribution $\mathcal{D}$ satisfying $\Pr_{(x,y)\sim\mathcal{D}}[y\,\langle w^*, x \rangle \geq 1] = 1$ and $\Pr_{(x,y)\sim\mathcal{D}}[\|x\| \leq R] = 1$. Let $\gamma$ be the maximum margin over all weight vectors classifying the uncorrupted data, and let $\mathcal{F}_{\mathsf{adv}} = \{\mathsf{patch}\,(x) \ : \ \|\mathsf{patch}\,(x) - x\| \leq \gamma\}$.*

*If $S_{\mathsf{clean}}$ consists of at least $\Omega\left(\varepsilon_{\mathsf{clean}}^{-2}\left(\gamma^{-2}R^2 + \log\left(1/\delta\right)\right)\right)$ i.i.d examples drawn from $\mathcal{D}$ and if $S_{\mathsf{adv}}$ consists of at least $\Omega\left(\varepsilon_{\mathsf{adv}}^{-2}\left(\gamma^{-2}R^2 + \log\left(1/\delta\right)\right)\right)$ i.i.d examples drawn from $\mathcal{D}|h^*(x) \neq t$, then we have:*

$$\min_{w \ : \ \|w\| \leq \gamma^{-1}} \frac{1}{|S|} \sum_{(x,y) \in S} \mathbb{1}\left\{y\,\langle w, x \rangle < 1\right\} > 0$$

*In other words, assuming there exists a margin $\gamma$ and a 0-loss classifier, empirical risk minimization of margin-loss with a norm constraint fails to find a 0-loss classifier on a sufficiently contaminated training set.*

## 2.3 Memorization Capacity and Backdoor Attacks

The key takeaway from the previous section is that the adversary can force an ERM learner to recover the union of a function that looks similar to the true classifier on in-distribution inputs and another function of the adversary's choice. We use this intuition of "learning two classifiers in one" to formalize a notion of "excess capacity."

To this end, we define the *memorization capacity* of a class and a domain.

**Definition 7** (Memorization Capacity). *Suppose we are in a setting where we are learning a hypothesis class $\mathcal{H}$ over a domain $\mathcal{X}$ under distribution $\mathcal{D}$.*

*We say we can* memorize $k$ irrelevant *sets from a family $\mathcal{C}$ atop a fixed $h^*$ if we can find $k$ pairwise disjoint nonempty sets $X_1, \ldots, X_k$ from a family of subsets of the domain $\mathcal{C}$ such that for all $b \in \{\pm 1\}^k$, there exists a classifier $\widehat{h} \in \mathcal{H}$ satisfying the below:*

- *For all $x \in X_i$, we have $\widehat{h}(x) = b_i$.*

- $\Pr_{x \sim \mathcal{D}}\left[\widehat{h}(x) = h^*(x)\right] = 1.$

*We define $\mathsf{mcap}_{\mathcal{X}, \mathcal{D}}\,(h, \mathcal{H}, \mathcal{C})$ to be the maximum number of sets from $\mathcal{C}$ we can memorize for a fixed $h$ belonging to a hypothesis class $\mathcal{H}$. We define $\mathsf{mcap}_{\mathcal{X}, \mathcal{D}}\,(h, \mathcal{H}) = \mathsf{mcap}_{\mathcal{X}, \mathcal{D}}\,(h, \mathcal{H}, \mathcal{B}_{\mathcal{X}})$ to be the maximum number of sets from $\mathcal{B}_{\mathcal{X}}$ we can memorize for a fixed $h$, where $\mathcal{B}_{\mathcal{X}}$ is the family of all non-empty measurable subsets of $\mathcal{X}$. Finally, we define $\mathsf{mcap}_{\mathcal{X}, \mathcal{D}}\,(\mathcal{H}) := \sup_{h \in \mathcal{H}} \mathsf{mcap}_{\mathcal{X}, \mathcal{D}}\,(h, \mathcal{H})$.*

Intuitively, the memorization capacity captures the number of additional irrelevant (with respect to $\mathcal{D}$) sets that can be memorized atop a true classifier.

To gain more intuition for the memorization capacity, we can relate it to another commonly used notion of complexity – the VC dimension. Specifically, we have the following lemma.

**Lemma 8** ((Appendix Lemma 23)). *We have* $0 \leq \mathsf{mcap}_{\mathcal{X},\mathcal{D}}(\mathcal{H}) \leq \mathsf{VC}(\mathcal{H})$.

Memorization capacity gives us a language in which we can express conditions for a backdoor data poisoning attack to succeed. Specifically, we have the following general result.

**Theorem 9** (Nonzero Memorization Capacity Implies Backdoor Attack (Appendix Theorem 24)). *Pick a target label $t \in \pm 1$. Suppose we have a hypothesis class $\mathcal{H}$, a target function $h^*$, a domain $\mathcal{X}$, a data distribution $\mathcal{D}$, and a class of patch functions $\mathcal{F}_{\mathsf{adv}}$. Define:*

$$\mathcal{C}(\mathcal{F}_{\mathsf{adv}}(h^*)) \coloneqq \{\mathsf{patch}\,(\mathsf{Supp}\,(\mathcal{D}|h^*(x) \neq t)) \ : \ \mathsf{patch} \in \mathcal{F}_{\mathsf{adv}}\}$$

*Now, suppose that* $\mathsf{mcap}_{\mathcal{X},\mathcal{D}}(h^*, \mathcal{H}, \mathcal{C}(\mathcal{F}_{\mathsf{adv}}(h^*))) \geq 1$. *Then, there exists a function* $\mathsf{patch} \in \mathcal{F}_{\mathsf{adv}}$ *for which the adversary can draw a set $S_{\mathsf{adv}}$ consisting of $m = \Omega\left(\varepsilon_{\mathsf{adv}}^{-1}\left(\mathsf{VC}(\mathcal{H}) + \log\left(1/\delta\right)\right)\right)$ i.i.d samples from $\mathcal{D}|h^*(x) \neq t$ such that with probability at least $1 - \delta$ over the draws of $S_{\mathsf{adv}}$, the adversary achieves the objectives of Problem 2, regardless of the number of samples the learner draws from $\mathcal{D}$ for $S_{\mathsf{clean}}$.*

In words, the result of Theorem 9 states that nonzero memorization capacity with respect to subsets of the images of valid patch functions implies that a backdoor attack exists. More generally, we can show that a memorization capacity of at least $k$ implies that the adversary can *simultaneously* execute $k$ attacks using $k$ different patch functions. In practice, this could amount to, for instance, selecting $k$ different triggers for an image and correlating them with various desired outputs. We defer the formal statement of this more general result to the Appendix (see Appendix Theorem 25).

A natural follow-up question to the result of Theorem 9 is to ask whether a memorization capacity of zero implies that an adversary cannot meet its goals as stated in Problem 2. Theorem 10 answers this affirmatively.

**Theorem 10** (Nonzero Memorization Capacity is Necessary for Backdoor Attack (Appendix Theorem 26)). *Let $\mathcal{C}(\mathcal{F}_{\mathsf{adv}}(h^*))$ be defined the same as in Theorem 9. Suppose we have a hypothesis class $\mathcal{H}$ over a domain $\mathcal{X}$, a true classifier $h^*$, data distribution $\mathcal{D}$, and a perturbation class $\mathcal{F}_{\mathsf{adv}}$. If $\mathsf{mcap}_{\mathcal{X},\mathcal{D}}(h^*, \mathcal{H}, \mathcal{C}(\mathcal{F}_{\mathsf{adv}}(h^*))) = 0$, then the adversary cannot successfully construct a backdoor data poisoning attack as per the conditions of Problem 2.*

### 2.3.1 Examples
We now use our notion of memorization capacity to examine the vulnerability of several natural learning problems to backdoor data poisoning attacks.

**Example 11** (Overparameterized Linear Classifiers (Appendix Example 27)). *Recall the result from the previous section, where we took $\mathcal{X} = \mathbb{R}^d$, $\mathcal{H}_d$ to be the set of linear classifiers in $\mathbb{R}^d$, and let $\mathcal{D}$ be a distribution over a radius-$R$ subset of an $s$-dimensional subspace $P$. We also assume that the true labeler $h^*$ achieves margin $\gamma$.*

*If we set $\mathcal{F}_{\mathsf{adv}} = \{\mathsf{patch}\,(x) \ : \ \mathsf{patch}\,(x) = x + \eta, \eta \in \mathbb{R}^d\}$, then we have $\mathsf{mcap}_{\mathcal{X},\mathcal{D}}(h^*, \mathcal{H}_d, \mathcal{C}(\mathcal{F}_{\mathsf{adv}}(h^*))) \geq d - s$.*

**Example 12** (Linear Classifiers Over Convex Bodies (Appendix Example 28)). *Let $\mathcal{H}$ be the set of origin-containing halfspaces. Fix an origin-containing halfspace $h^*$ with weight vector $w^*$. Let $\mathcal{X}'$ be a closed compact convex set, let $\mathcal{X} = \mathcal{X}' \setminus \{x \ : \ \langle w^*, x \rangle = 0\}$, and let $\mathcal{D}$ be any probability measure over $\mathcal{X}$ that assigns nonzero measure to every $\ell_2$ ball of nonzero radius contained in $\mathcal{X}$ and satisfies the relation $\mu_{\mathcal{D}}(Y) = 0 \iff \mathsf{Vol}_d(Y) = 0$ for all $Y \subset \mathcal{X}$. Then, $\mathsf{mcap}_{\mathcal{X},\mathcal{D}}(h^*, \mathcal{H}) = 0$.*

Given these examples, it is natural to wonder whether memorization capacity can be greater than $0$ when the support of $\mathcal{D}$ is the entire space $\mathcal{X}$. The following example shows this indeed can be the case.

**Example 13** (Sign Changes (Appendix Example 29)). *Let $\mathcal{X} = [0, 1]$, $\mathcal{D} = \mathsf{Unif}(\mathcal{X})$ and $\mathcal{H}_k$ be the class of functions admitting at most $k$ sign-changes. Specifically, $\mathcal{H}_k$ consists of functions $h$ for which we can find pairwise disjoint, continuous intervals $I_1, \ldots, I_{k+1}$ such that:*

- *For all $i < j$ and for all $x \in I_i, y \in I_j$, we have $x < y$.*

- $\bigcup_{i=1}^{k+1} I_i = \mathcal{X}$.

- $h(I_i) = -h(I_{i+1})$, *for all $i \in [k]$.*

*Suppose the learner is learning $\mathcal{H}_s$ for unknown $s$ using $\mathcal{H}_d$, where $s \leq d + 2$. For all $h^* \in \mathcal{H}_s$, we have $\mathsf{mcap}_{\mathcal{X}, \mathcal{D}} (h^*, \mathcal{H}_d) \geq \lfloor (d-s)/2 \rfloor$.*

## 3  Algorithmic Considerations

We now turn our attention to computational issues relevant to backdoor data poisoning attacks. Throughout the rest of this section, define the adversarial loss:

$$\mathcal{L}_{\mathcal{F}_{\mathsf{adv}}(h^*)}(\widehat{h}, S) := \mathop{\mathbb{E}}_{(x,y) \sim S} \left[ \sup_{\mathsf{patch} \in \mathcal{F}_{\mathsf{adv}}(h^*)} \mathbb{1}\left\{ \widehat{h}(\mathsf{patch}(x)) \neq y \right\} \right]$$

In a slight overload of notation, let $\mathcal{L}^{\mathcal{H}}_{\mathcal{F}_{\mathsf{adv}}(h^*)}$ denote the robust loss class of $\mathcal{H}$ with the perturbation sets generated by $\mathcal{F}_{\mathsf{adv}}(h^*)$:

$$\mathcal{L}^{\mathcal{H}}_{\mathcal{F}_{\mathsf{adv}}(h^*)} := \left\{ (x, y) \mapsto \sup_{\mathsf{patch} \in \mathcal{F}_{\mathsf{adv}}(h^*)} \mathbb{1}\left\{ \widehat{h}(\mathsf{patch}(x)) \neq y \right\} : \widehat{h} \in \mathcal{H} \right\}$$

Then, assume that $\mathsf{VC}\left( \mathcal{L}^{\mathcal{H}}_{\mathcal{F}_{\mathsf{adv}}(h^*)} \right)$ is finite[3]. Finally, assume that the perturbation set $\mathcal{F}_{\mathsf{adv}}$ is the same as that consistent with the ground-truth classifier $h^*$. In other words, once $h^*$ is selected, then we reveal to both the learner and the adversary the sets $\mathcal{F}_{\mathsf{adv}}(h^*)$; thus, the learner equates $\mathcal{F}_{\mathsf{adv}}$ and $\mathcal{F}_{\mathsf{adv}}(h^*)$. Hence, although $h^*$ is not known to the learner, $\mathcal{F}_{\mathsf{adv}}(h^*)$ is. As an example of a natural scenario in which such an assumption holds, consider the case where $h^*$ is some large-margin classifier and $\mathcal{F}_{\mathsf{adv}}$ consists of short additive perturbations. This subsumes the setting where $h^*$ is some image classifier and $\mathcal{F}_{\mathsf{adv}}$ consists of test-time adversarial perturbations which don't impact the true classifications of the source images.

### 3.1  Certifying the Existence of Backdoors

The assumption that $\mathcal{F}_{\mathsf{adv}} = \mathcal{F}_{\mathsf{adv}}(h^*)$ gives the learner enough information to minimize $\mathcal{L}_{\mathcal{F}_{\mathsf{adv}}(h^*)}(\widehat{h}, S)$ on a finite training set $S$ over $\widehat{h} \in \mathcal{H}$[4]; the assumption that $\mathsf{VC}\left( \mathcal{L}^{\mathcal{H}}_{\mathcal{F}_{\mathsf{adv}}(h^*)} \right) < \infty$ yields that the learner recovers a classifier that has low robust loss as per uniform convergence. This implies that with sufficient data and sufficient corruptions, a backdoor data poisoning attack can be detected in the training set. We formalize this below.

**Theorem 14** (Certifying Backdoor Existence (Appendix Theorem 30)). *Suppose that the learner can calculate and minimize:*

$$\mathcal{L}_{\mathcal{F}_{\mathsf{adv}}(h^*)}(\widehat{h}, S) = \mathop{\mathbb{E}}_{(x,y) \sim S} \left[ \sup_{\mathsf{patch} \in \mathcal{F}_{\mathsf{adv}}(h^*)} \mathbb{1}\left\{ \widehat{h}(\mathsf{patch}(x)) \neq y \right\} \right]$$

*over a finite set $S$ and $\widehat{h} \in \mathcal{H}$.*

*If the VC dimension of the loss class $\mathcal{L}^{\mathcal{H}}_{\mathcal{F}_{\mathsf{adv}}(h^*)}$ is finite, then there exists an algorithm using $O\left( \varepsilon_{\mathsf{clean}}^{-2} \left( \mathsf{VC}\left( \mathcal{L}_{\mathcal{F}_{\mathsf{adv}}(h^*)} \right) + \log(1/\delta) \right) \right)$ samples that allows the learner to defeat the adversary through learning a backdoor-robust classifier or by rejecting the training set as being corrupted, with probability $1 - \delta$.*

---

[3]It is shown in [21] that there exist classes $\mathcal{H}$ and corresponding adversarial loss classes $\mathcal{L}_{\mathcal{F}_{\mathsf{adv}}(h^*)}$ for which $\mathsf{VC}(\mathcal{H}) < \infty$ but $\mathsf{VC}\left( \mathcal{L}^{\mathcal{H}}_{\mathcal{F}_{\mathsf{adv}}(h^*)} \right) = \infty$. Nonetheless, there are a variety of natural scenarios in which we have $\mathsf{VC}(\mathcal{H}), \mathsf{VC}\left( \mathcal{L}^{\mathcal{H}}_{\mathcal{F}_{\mathsf{adv}}(h^*)} \right) < \infty$; for example, in the case of linear classifiers in $\mathbb{R}^d$ and for closed, convex, origin-symmetric, additive perturbation sets, we have $\mathsf{VC}(\mathcal{H}), \mathsf{VC}\left( \mathcal{L}^{\mathcal{H}}_{\mathcal{F}_{\mathsf{adv}}(h^*)} \right) \leq d + 1$ (see [26] [14]).

[4]However, minimizing $\mathcal{L}_{\mathcal{F}_{\mathsf{adv}}(h^*)}$ might be computationally intractable in several scenarios.

See Algorithm A.1 in the Appendix for the pseudocode of an algorithm witnessing the statement of Theorem 14.

Our result fleshes out and validates the approach implied by [31], where the authors use data augmentation to robustly learn in the presence of backdoors. Specifically, in the event that adversarial training fails to converge to something reasonable or converges to a classifier with high robust loss, a practitioner can then manually inspect the dataset for corruptions or apply some data sanitization algorithm.

### 3.1.1 Numerical Trials

To exemplify such a workflow, we implement adversarial training in a backdoor data poisoning setting. Specifically, we select a target label, inject a varying fraction of poisoned examples into the MNIST dataset (see [2]), and estimate the robust training and test loss for each choice of $\alpha$. Our results demonstrate that in this setting, the training robust loss indeed increases with the fraction of corrupted data $\alpha$; moreover, the classifiers obtained with low training robust loss enjoy a low test-time robust loss. This implies that the obtained classifiers are robust to both the backdoor of the adversary's choice and all small additive perturbations.

For a more detailed description of our methodology, setup, and results, please see Appendix Section B.

### 3.2 Filtering versus Generalization

We now show that two related problems we call *backdoor filtering* and *robust generalization* are nearly statistically equivalent; computational equivalence follows if there exists an efficient algorithm to minimize $\mathcal{L}_{\mathcal{F}_{\text{adv}}(h^*)}$ on a finite training set. We first define these two problems below (Problems 15 and 16).

**Problem 15** (Backdoor Filtering). *Given a training set $S = S_{\text{clean}} \cup S_{\text{adv}}$ such that $|S_{\text{clean}}| \geq \Omega\left(\text{poly}\left(\varepsilon^{-1}, \log\left(1/\delta\right), \text{VC}\left(\mathcal{L}_{\mathcal{F}_{\text{adv}}(h^*)}\right)\right)\right)$, return a subset $S' \subseteq S$ such that the solution to the optimization $\widehat{h} := \text{argmin}_{h \in \mathcal{H}} \mathcal{L}_{\mathcal{F}_{\text{adv}}(h^*)}(h, S')$ satisfies $\mathcal{L}_{\mathcal{F}_{\text{adv}}(h^*)}(h, \mathcal{D}) \lesssim \varepsilon_{\text{clean}}$ with probability $1 - \delta$.*

Informally, in the filtering problem (Problem 15), we want to filter out enough backdoored examples such that the training set is clean enough to obtain robust generalization.

**Problem 16** (Robust Generalization). *Given a training set $S = S_{\text{clean}} \cup S_{\text{adv}}$ such that $|S_{\text{clean}}| \geq \Omega\left(\text{poly}\left(\varepsilon^{-1}, \log\left(1/\delta\right), \text{VC}\left(\mathcal{L}_{\mathcal{F}_{\text{adv}}(h^*)}\right)\right)\right)$, return a classifier $\widehat{h}$ satisfies $\mathcal{L}_{\mathcal{F}_{\text{adv}}(h^*)}\widehat{h}, \mathcal{D} \leq \varepsilon_{\text{clean}}$ with probability $1 - \delta$.*

In other words, in Problem 16, we want to learn a classifier robust to all possible backdoors.

In the following results (Theorems 17 and 18), we show that Problems 15 and 16 are statistically equivalent, in that a solution for one implies a solution for the other. Specifically, we can write the below.

**Theorem 17** (Filtering Implies Generalization (Appendix Theorem 31)). *Let $\alpha \leq 1/3$ and $\varepsilon_{\text{clean}} \leq 1/10$.*

*Suppose we have a training set $S = S_{\text{clean}} \cup S_{\text{adv}}$ such that $|S_{\text{clean}}| = \Omega\left(\varepsilon_{\text{clean}}^{-2}\left(\text{VC}\left(\mathcal{L}_{\mathcal{F}_{\text{adv}}(h^*)}\right) + \log\left(1/\delta\right)\right)\right)$ and $|S_{\text{adv}}| \leq \alpha \cdot (|S_{\text{adv}}| + |S_{\text{clean}}|)$. If there exists an algorithm that given $S$ can find a subset $S' = S'_{\text{clean}} \cup S'_{\text{adv}}$ satisfying $|S'_{\text{clean}}|/|S_{\text{clean}}| \geq 1 - \varepsilon_{\text{clean}}$ and $\min_{h \in \mathcal{H}} \mathcal{L}_{\mathcal{F}_{\text{adv}}(h^*)}(h, S') \lesssim \varepsilon_{\text{clean}}$, then there exists an algorithm such that given $S$ returns a function $\widehat{h}$ satisfying $\mathcal{L}_{\mathcal{F}_{\text{adv}}(h^*)}(\widehat{h}, \mathcal{D}) \lesssim \varepsilon_{\text{clean}}$ with probability $1 - \delta$.*

See Algorithm A.2 in the Appendix for the pseudocode of an algorithm witnessing the theorem statement.

**Theorem 18** (Generalization Implies Filtering (Appendix Theorem 33)). *Set $\varepsilon_{\text{clean}} \leq 1/10$ and $\alpha \leq 1/6$.*

*If there exists an algorithm that, given at most a $2\alpha$ fraction of outliers in the training set, can output a hypothesis satisfying $\mathcal{L}_{\mathcal{F}_{\text{adv}}(h^*)}(\widehat{h}, \mathcal{D}) \leq \varepsilon_{\text{clean}}$ with probability $1 - \delta$ over the draw of the training set, then there exists an algorithm that given a training set $S = S_{\text{clean}} \cup S_{\text{adv}}$ satisfying*

$|S_{\mathsf{clean}}| \geq \Omega\left(\varepsilon_{\mathsf{clean}}^{-2}\left(\mathsf{VC}\left(\mathcal{L}_{\mathcal{F}_{\mathsf{adv}(h^*)}}\right) + \log\left(1/\delta\right)\right)\right)$ *outputs a subset* $S' \subseteq S$ *with the property that* $\mathcal{L}_{\mathcal{F}_{\mathsf{adv}(h^*)}}\left(\mathrm{argmin}_{h \in \mathcal{H}}\mathcal{L}_{\mathcal{F}_{\mathsf{adv}(h^*)}}\left(h, S'\right), \mathcal{D}\right) \lesssim \varepsilon_{\mathsf{clean}}$ *with probability* $1 - 7\delta$.

See Algorithm A.3 in the Appendix for the pseudocode of an algorithm witnessing Theorem 18. Note that there is a factor-2 separation between the values of $\alpha$ used in the filtering and generalizing routines above; this is a limitation of our current analysis.

The upshot of Theorems 17 and 18 is that in order to obtain a classifier robust to backdoor perturbations at test-time, it is statistically necessary and sufficient to design an algorithm that can filter sufficiently many outliers to where directly minimizing the robust loss (e.g., adversarial training) yields a generalizing classifier. Furthermore, computational equivalence holds in the case where minimizing the robust loss on the training set can be done efficiently (such as in the case of linear separators with closed, convex, bounded, origin-symmetric perturbation sets – see [26]). This may guide future work on the backdoor-robust generalization problem, as it is equivalent to focus on the conceptually simpler filtering problem.

# 4 Related Works

Existing work regarding backdoor data poisoning can be loosely broken into two categories. For a more general survey of backdoor attacks, please see the work of [25].

**Attacks**  To the best of our knowledge, the first work to empirically demonstrate the existence of backdoor poisoning attacks is that of [10]. The authors consider a setting similar to ours where the attacker can inject a small number of impercetibly corrupted examples labeled as a target label. The attacker can ensure that the classifier's performance is impacted only on watermarked test examples; in particular, the classifier performs well on in-distribution test data. Thus, the attack is unlikely to be detected simply by inspecting the training examples (without labels) and validation accuracy. The work of [9] and [19] explores a similar setting.

The work of [30] discusses theoretical aspects of backdoor poisoning attacks in a federated learning scenario. Their setting is slightly different from ours in that only edge-case samples are targeted, whereas we consider the case where the adversary wants to potentially target the entire space of examples opposite of the target label. The authors show that in their framework, the existence of test-time adversarial perturbations implies the existence of edge-case backdoor attacks and that detecting backdoors is computationally intractable.

Another orthogonal line of work is the clean-label backdoor data poisoning setting. Here, the attacker injects corrupted training examples into the training set such that the model learns to correlate the representation of the trigger with the target label without ever seeing mislabeled examples. The work of [27] and [23] give empirically successful constructions of such an attack. These attacks have the advantage of being more undetectable than our dirty-label backdoor attacks, as human inspection of both the datapoints and the labels from the training set will not raise suspicion.

Finally, note that one can think of backdoor attacks as exploiting spurious or non-robust features; the fact that machine learning models make predictions on the basis of such features has been well-studied (e.g. see [6], [20], [32]).

**Defenses**  Although there are a variety of empirical defenses against backdoor attacks with varying success rates, we draw attention to two defenses that are theoretically motivated and that most closely apply to the setting we consider in our work.

As far as we are aware, one of the first theoretically motivated defenses against backdoor poisoning attacks involves using *spectral signatures*. Spectral signatures ([17]) relies on the fact that outliers necessarily corrupt higher-order moments of the empirical distribution, especially in the feature space. Thus, to find outliers, one can estimate class means and covariances and filter the points most correlated with high-variance projections of the empirical distribution in the feature space. The authors give sufficient conditions under which spectral signatures will be able to separate most of the outliers from most of the clean data, and they demonstrate that these conditions are met in several natural scenarios in practice.

Another defense with some provable backing is *Iterative Trimmed Loss Minimization* (ITLM), which was first used against backdoor attacks by [22]. ITLM is an algorithmic framework motivated by the idea that the value of the loss function on the set of clean points may be lower than that on the set

of corrupted points. Thus, an ITLM-based procedure selects a low-loss subset of the training data and performs a model update step on this subset. This alternating minimization is repeated until the model loss is sufficiently small. The heuristic behind ITLM holds in practice, as per the evaluations from [22].

**Memorization of Training Data**    The work of [8] and [24] discuss the ability of neural networks to memorize their training data. Specifically, the work of [8] empirically discusses how memorization plays into the learning dynamics of neural networks via fitting random labels. The work of [24] experimentally validates the "long tail theory", which posits that data distributions in practice tend to have a large fraction of their mass allocated to "atypical" examples; thus, the memorization of these rare examples is actually necessary for generalization.

Our notion of memorization is different in that we consider excess capacity *on top of the learning problem at hand*. In other words, we require that there exist a classifier in the hypothesis class that behaves correctly on on-distribution data in addition to memorizing specially curated off-distribution data.

## 5   Conclusions and Future Work

**Conclusions**    We gave a framework under which backdoor data poisoning attacks can be studied. We then showed that, under this framework, a formal notion of excess capacity present in the learning problem is necessary and sufficient for the existence of a backdoor attack. Finally, in the algorithmic setting, we showed that under certain assumptions, adversarial training can detect the presence of backdoors and that filtering backdoors from a training set is equivalent to learning a backdoor-robust classifier.

**Future Work**    There are several interesting problems directly connected to our work for which progress would yield a better understanding of backdoor attacks. Perhaps the most important is to find problems for which there simultaneously exist efficient backdoor filtering algorithms and efficient adversarial training algorithms. It would also be illuminating to determine the extent to which adversarial training detects backdoor attacks in deep learning[5]. Finally, we believe that our notion of memorization capacity can find applications beyond the scope of this work. It would be particularly interesting to see if memorization capacity has applications to explaining robustness or lack thereof to test-time adversarial perturbations.

**Societal Impacts**    Defenses against backdoor attacks may impede the functionality of several privacy-preserving applications. Most notably, the Fawkes system (see [28]) relies on a backdoor data poisoning attack to preserve its users' privacy, and such a system could be compromised if it were known how to reliably defend against backdoor data poisoning attacks in such a setting.

**Acknowledgments**    This work was supported in part by the National Science Foundation under grant CCF-1815011 and by the Defense Advanced Research Projects Agency under cooperative agreement HR00112020003. The views expressed in this work do not necessarily reflect the position or the policy of the Government and no official endorsement should be inferred. Approved for public release; distribution is unlimited.

NSM thanks Surbhi Goel for suggesting the experiments run in the paper.

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
