## A Restatement of Theorems and Full Proofs

In this section, we will restate our main results and give full proofs.

**Theorem 19** (Existence of Backdoor Data Poisoning Attack (Theorem 3)). *Let $\mathcal{F}_{\mathsf{adv}}$ be some family of patch functions such that for all $i \in U$, $\Pr_{x \sim \mathcal{D}} [v_i(\mathsf{patch}\,(x)) = v_i(x)] = 1$, there exists at least one $j \in [\dim(\mathcal{V})] \setminus U$ such that $\Pr_{x \sim \mathcal{D}} [v_j(\mathsf{patch}\,(x)) \neq 0] = 1$, and for all $j \in [\dim(\mathcal{V})]$, we either have $\Pr_{x \sim \mathcal{D}} [v_j(\mathsf{patch}\,(x)) \geq 0] = 1$ or $\Pr_{x \sim \mathcal{D}} [v_j(\mathsf{patch}\,(x)) \leq 0] = 1$.*

*Fix any target label $t \in \{\pm 1\}$. Draw a training set $S_{\mathsf{clean}}$ of size at least $m_0 := \Omega\left(\varepsilon_{\mathsf{clean}}^{-1} (\mathsf{VC}\,(\mathcal{H}) + \log(1/\delta))\right)$. Then, draw a backdoor training set $S_{\mathsf{adv}}$ of size at least $m_1 := \Omega\left(\varepsilon_{\mathsf{adv}}^{-1} (\mathsf{VC}\,(\mathcal{H}) + \log(1/\delta))\right)$ of the form $(x, t)$ where $x \sim \mathsf{patch}\,(\mathcal{D} | h^*(x) \neq t)$.*

*With probability at least $1 - \delta$, empirical risk minimization on the training set $S := S_{\mathsf{clean}} \cup S_{\mathsf{adv}}$ yields a classifier $\widehat{h}$ satisfying the success conditions for Problem 2.*

*Proof.* We first show that our choice of $\mathcal{F}_{\mathsf{adv}}$ is valid. Observe that for our choice of $h^*$, we have that with probability 1:

$$
\begin{aligned}
h^*\,(\mathsf{patch}\,(x)) &= \mathrm{sign}\left(\sum_{i \in U} a_i \cdot v_i(\mathsf{patch}\,(x))\right) \\
&= \mathrm{sign}\left(\sum_{i \in U} a_i \cdot v_i(x)\right) \quad , \text{ since } \Pr_{x \sim \mathcal{D}} [v_i(\mathsf{patch}\,(x)) = v_i(x)] = 1 \\
&= h^*\,(x)
\end{aligned}
$$

as desired.

Next, we pick a patch function from $\mathcal{F}_{\mathsf{adv}}$ and construct our stamped training set. We will show that there exists a classifier in $\mathcal{H}$ achieving $0$ training error on the corrupted training set $S_{\mathsf{clean}} \cup S_{\mathsf{adv}}$. Without loss of generality, assume that $\Pr_{x \sim \mathcal{D}} [v_i(\mathsf{patch}\,(x)) \geq 0] = 1$, for all $i \in [\dim(\mathcal{V})]$ (as negating the basis vector doesn't affect the space spanned).

Specifically, we will show that the following function achieves $0$ training error on the corrupted training set $S = S_{\mathsf{clean}} \cup S_{\mathsf{adv}}$ (let $a_i$, $i \in U$ denote the coefficients corresponding to $h^*$, let $R = \max_{x \in S_{\mathsf{clean}}} |v^*(x)|$, and let $j$ be any index in $[\dim(\mathcal{V})] \setminus U$:

$$
\widehat{h}(x) := \mathrm{sign}\left(\sum_{i \in U} a_i \cdot v_i(x) + t \cdot \frac{2R}{\min_{x \in S_{\mathsf{adv}}} v_j(\mathsf{patch}\,(x))} \cdot v_j(x)\right)
$$

We first evaluate $\widehat{h}(x)$ on $x \sim \mathcal{D}$. We have, with probability 1:

$$
\begin{aligned}
\widehat{h}(x) &= \mathrm{sign}\left(\sum_{i \in U} a_i \cdot v_i(x) + t \cdot \frac{2R}{\min_{x \in S_{\mathsf{adv}}} v_j(\mathsf{patch}\,(x))} \cdot v_j(x)\right) \\
&= \mathrm{sign}\left(\sum_{i \in U} a_i \cdot v_i(x)\right) \quad , \text{ since } \Pr_{x \sim \mathcal{D}} [v_j(x) = 0] = 1 \\
&= h^*(x)
\end{aligned}
$$

We now evaluate $\widehat{h}(\mathsf{patch}\,(x))$ on $x \sim \mathcal{D} | h^*(x) \neq t$. We have, with probability 1:

$$
\begin{aligned}
\widehat{h}(x) &= \mathrm{sign}\left(\sum_{i \in U} a_i \cdot v_i(\mathsf{patch}\,(x)) + t \cdot \frac{2R}{\min_{x \in S_{\mathsf{adv}}} v_j(\mathsf{patch}\,(x))} \cdot v_j(\mathsf{patch}\,(x))\right) \\
&= \mathrm{sign}\left(\sum_{i \in U} a_i \cdot v_i(x) + t \cdot \frac{2R}{\min_{x \in S_{\mathsf{adv}}} v_j(\mathsf{patch}\,(x))} \cdot v_j(\mathsf{patch}\,(x))\right) \\
&= \mathrm{sign}\left(v^*(x) + t \cdot \frac{2R}{\min_{x \in S_{\mathsf{adv}}} v_j(\mathsf{patch}\,(x))} \cdot v_j(\mathsf{patch}\,(x))\right)
\end{aligned}
$$

We now have to show that regardless of the $x \in \mathcal{X}$ we pick, we have that the above expression evaluates to $t$ with probability 1. We have two cases. In the first case, $h^*(x) = 1$, which implies that $v^*(x) \leq R$. Since $t = -1$, we write:

$$\widehat{h}(\mathsf{patch}\,(x)) = \mathrm{sign}\left(v^*(x) + t \cdot \frac{2R}{\min_{x \in S_{\mathsf{adv}}} v_j(\mathsf{patch}\,(x))} \cdot v_j(\mathsf{patch}\,(x))\right)$$

$$= \mathrm{sign}\left(R - \frac{2R}{\min_{x \in S_{\mathsf{adv}}} v_j(\mathsf{patch}\,(x))} \cdot v_j(\mathsf{patch}\,(x))\right)$$

$$= \mathrm{sign}\left(1 - \frac{2}{\min_{x \in S_{\mathsf{adv}}} v_j(\mathsf{patch}\,(x))} \cdot v_j(\mathsf{patch}\,(x))\right)$$

$$= -1$$

where the last line follows from the fact that $1 - \frac{2}{\min_{x \in S_{\mathsf{adv}}} v_j(\mathsf{patch}(x))} \cdot v_j(\mathsf{patch}\,(x)) \leq -1$.

The proof in the event where $t = 1$ follows similarly. It thus follows that there exists a function $\widehat{h} \in \mathcal{H}$ obtaining 0 error on $S_{\mathsf{clean}} \cup S_{\mathsf{adv}}$. The desired result immediately follows from uniform convergence (we have a 0-error classifier over $S_{\mathsf{clean}} \sim \mathcal{D}$ and a 0-error classifier over $S_{\mathsf{adv}} \sim \mathsf{patch}\,(\mathcal{D}|h^*(x) \neq t)$, so with probability $1 - 2\delta$, we have error at most $\varepsilon_{\mathsf{clean}}$ on the clean distribution and error at most $\varepsilon_{\mathsf{adv}}$ on the adversarial distribution). $\qquad\square$

**Corollary 20** (Overparameterized Linear Classifier (Corollary 4)). *Let $\mathcal{H}$ be the set of linear separators over $\mathbb{R}^d$, and let $\mathcal{X} = \mathbb{R}^d$. Let $\mathcal{D}$ be some distribution over an $s$-dimensional subspace of $\mathbb{R}^d$ where $s < d$, so with probability 1, we can write $x \sim \mathcal{D}$ as $Az$ for some $A \in \mathbb{R}^{d \times s}$ and for $z \in \mathbb{R}^s$. Let $\mathcal{F}_{\mathsf{adv}} = \{\mathsf{patch}\,(x) \; : \; \mathsf{patch}\,(x) + \eta, \eta \perp \mathsf{Span}\,(A)\}$, and draw some patch function $\mathsf{patch} \in \mathcal{F}_{\mathsf{adv}}$.*

*Fix any target label $t \in \{\pm 1\}$. Draw a training set $S_{\mathsf{clean}}$ of size at least $m_0 := \Omega\left(\varepsilon_{\mathsf{clean}}^{-1}\left(\mathsf{VC}\,(\mathcal{H}) + \log\,(1/\delta)\right)\right)$. Then, draw a backdoor training set $S_{\mathsf{adv}}$ of size at least $m_1 := \Omega\left(\varepsilon_{\mathsf{adv}}^{-1}\left(\mathsf{VC}\,(\mathcal{H}) + \log\,(1/\delta)\right)\right)$ of the form $(x, t)$ where $x \sim (\mathcal{D}|h^*(x) \neq t) + \eta$.*

*With probability at least $1 - \delta$, empirical risk minimization on the training set $S_{\mathsf{clean}} \cup S_{\mathsf{adv}}$ yields a classifier $\widehat{h}$ satisfying the success conditions for Problem 2.*

*Proof.* We will show that our problem setup is a special case of that considered in Theorem 3; then, we can apply that result as a black box.

Observe that the set of linear classifiers over $\mathbb{R}^d$ is a thresholded vector space with dimension $d$. Pick the basis $\{v_1, \ldots, v_s, \ldots, v_d\}$ such that $\{v_1, \ldots, v_s\}$ form a basis for the subspace $\mathsf{Span}\,(A)$ and $v_{s+1}, \ldots, v_d$ are some completion of the basis for the rest of $\mathbb{R}^d$.

Clearly, there is a size-$s$ set of indices $U \subset [d]$ such that for all $i \in U$, we have $\Pr_{x \sim \mathcal{D}}\left[v_i(x) \neq 0\right] > 0$. Without loss of generality, assume $U = [s]$.

Next, we need to show that for all $i \in U$, we have $v_i(\mathsf{patch}\,(x)) = 0$. Since we have $\eta \perp \mathsf{Span}\,(A)$, we have $v_i(\eta) = 0$ for all $i \in U$. Since the $v_i$ are also linear functions, we satisfy $v_i(Az + \eta) = 0$ for all $z \in \mathbb{R}^s$.

We now show that there is at least one $j \in [\dim\,(\mathcal{V})] \setminus U$ such that $\Pr_{x \sim \mathcal{D}}\left[v_j(\mathsf{patch}\,(x)) \neq 0\right] = 1$. Since $\eta \perp \mathsf{Span}\,(A)$, $\eta$ must be expressible as some nonzero linear combination of the vectors $v_j$; thus, taking the inner product with any such vector will result in a nonzero value.

Finally, we show that for all $j \in [\dim\,(\mathcal{V})] \setminus U$, we either have $\Pr_{x \sim \mathcal{D}}\left[v_j(\mathsf{patch}\,(x)) \geq 0\right] = 1$ or $\Pr_{x \sim \mathcal{D}}\left[v_j(\mathsf{patch}\,(x)) \leq 0\right] = 1$. Since $\eta$ is expressible as a linear combination of several such $v_j$, we can write:

$$\langle Az + \eta, v_j\rangle = \langle Az, v_j\rangle + \langle \eta, v_j\rangle$$

$$= 0 + \left\langle \sum_{j=s+1}^{d} a_j \cdot v_j, v_j\right\rangle$$

$$= a_j$$

which is clearly nonzero.

The result now follows from Theorem 3. $\qquad\square$

**Theorem 21** (Random direction is an adversarial watermark (Theorem 5))**.** *Consider the same setting used in Corollary 4, and set $\mathcal{F}_{\mathsf{adv}} = \left\{ \mathsf{patch} \; : \; \mathsf{patch}\,(x) = x + \eta, \eta \in \mathbb{R}^d \right\}$.*

*If $h^*$ achieves margin $\gamma$ and if the ambient dimension $d$ of the model satisfies $d \geq \Omega\left( s\log(s/\delta)/\gamma^2 \right)$, then an adversary can find a patch function such that with probability $1 - \delta$, a training set $S = S_{\mathsf{clean}} \cup S_{\mathsf{adv}}$ satisfying $|S_{\mathsf{clean}}| \geq \Omega\left( \varepsilon_{\mathsf{clean}}^{-1} \left( \mathsf{VC}\,(\mathcal{H}) + \log\left( 1/\delta \right) \right) \right)$ and $|S_{\mathsf{adv}}| \geq \Omega\left( \varepsilon_{\mathsf{clean}}^{-1} \left( \mathsf{VC}\,(\mathcal{H}) + \log\left( 1/\delta \right) \right) \right)$ yields a classifier $\widehat{h}$ satisfying the success conditions for Problem 2 while also satisfying $\underset{(x,y)\sim\mathcal{D}}{\mathbb{E}} \left[ \mathbb{1}\left\{ \widehat{h}(x) \neq y \right\} \right] \leq \varepsilon_{\mathsf{clean}}.$*

*This result holds true particularly when the adversary does not know $\mathsf{Supp}\,(\mathcal{D})$.*

*Proof.* We prove Theorem 5 in two parts. We first show that although the adversary doesn't know $\mathcal{F}_{\mathsf{adv}}(h^*)$, they can find $\mathsf{patch} \in \mathcal{F}_{\mathsf{adv}}(h^*)$ with high probability. We then invoke the result from Corollary 4.

Let $a_i$ denote the $i$th column of $A$. Next, draw $\eta$ from $\mathsf{Unif}\left( \mathbb{S}^{d-1} \right)$.

Recall that there exists a universal constant $C_0$ for which $\eta\sqrt{d}$ is $C_0$-subgaussian ([18]). Next, remember that if $\eta\sqrt{d}$ is $C_0$-subgaussian, then $\left\langle \eta\sqrt{d}, a_i \right\rangle$ has subgaussian constant $C_0 \left\| a_i \right\| = C_0$. Using classical subgaussian concentration inequalities, we arrive at the following:

$$\mathsf{Pr}\left[ \left| \left\langle \eta\sqrt{d}, a_i \right\rangle \right| \geq \frac{\varepsilon\sqrt{d}}{\sqrt{s}} \right] \leq 2\mathsf{exp}\left( -\frac{\varepsilon^2 d}{sC_0^2} \right)$$

$$\Rightarrow \mathsf{Pr}\left[ \text{for all } i \in [s], \; |\langle \eta, a_i \rangle| \leq \frac{\varepsilon}{\sqrt{s}} \right] \geq 1 - 2s \cdot \mathsf{exp}\left( -\frac{\varepsilon^2 d}{sC_0^2} \right)$$

$$\geq 1 - \frac{\delta}{2} \quad , \text{pick } d = \frac{C_0^2}{\varepsilon^2} \cdot s \cdot \log\left( \frac{4s}{\delta} \right)$$

Next, observe that if we have $|\langle \eta, a_i \rangle| \leq \varepsilon/\sqrt{s}$ for all $i \in [s]$, then we have:

$$\left\| A^T \eta \right\| = \sqrt{ \sum_{i=1}^{s} |\langle \eta, a_i \rangle|^2 }$$

$$\leq \sqrt{ \sum_{i=1}^{s} \frac{\varepsilon^2}{s} }$$

$$= \varepsilon$$

This implies that the norm of the component of the trigger in $\mathsf{Ker}\left( A^T \right)$ is at least $\sqrt{1 - \varepsilon^2} \geq 1 - \varepsilon$ from the Pythagorean Theorem.

Next, we substitute $\varepsilon = \gamma$. From this, we have that $\left\| A^T v \right\| \leq \gamma$ with probability $1 - \delta/2$, which implies that $h^*(x + \eta) = h^*(x)$ with probability $1 - \delta/2$ over the draws of $\eta$. This gives us that $\mathsf{patch}\,(x) = x + \eta \in \mathcal{F}_{\mathsf{adv}}(h^*)$ with probability $1 - \delta/2$ over the draws of $\eta$.

It is now easy to see that the result we want follows from a simple application of Corollary 4 using a failure probability of $\delta/2$, and we're done, where the final failure probability $1 - \delta$ follows from a union bound. $\qquad\square$

**Theorem 22** (Theorem 6)**.** *Consider some $h^*(x) = \mathsf{sign}\left( \langle w^*, x \rangle \right)$ and a data distribution $\mathcal{D}$ satisfying $\underset{(x,y)\sim\mathcal{D}}{\mathsf{Pr}} \left[ y \langle w^*, x \rangle \geq 1 \right] = 1$ and $\underset{(x,y)\sim\mathcal{D}}{\mathsf{Pr}} \left[ \|x\| \leq R \right] = 1$. Let $\gamma$ be the maximum margin over all weight vectors classifying the uncorrupted data, and let $\mathcal{F}_{\mathsf{adv}} = \left\{ \mathsf{patch}\,(x) \; : \; \|\mathsf{patch}\,(x) - x\| \leq \gamma \right\}.$*

*If $S_{\text{clean}}$ consists of at least $\Omega\left(\varepsilon_{\text{clean}}^{-2}\left(\gamma^{-2}R^2 + \log\left(1/\delta\right)\right)\right)$ i.i.d examples drawn from $\mathcal{D}$ and if $S_{\text{adv}}$ consists of at least $\Omega\left(\varepsilon_{\text{adv}}^{-2}\left(\gamma^{-2}R^2 + \log\left(1/\delta\right)\right)\right)$ i.i.d examples drawn from $\mathcal{D}|h^*(x) \neq t$, then we have:*

$$\min_{w \ : \ \|w\| \leq \gamma^{-1}} \frac{1}{|S|} \sum_{(x,y) \in S} \mathbb{1}\left\{y\left\langle w, x\right\rangle < 1\right\} > 0$$

*In other words, assuming there exists a margin $\gamma$ and a 0-loss classifier, empirical risk minimization of margin-loss with a norm constraint fails to find a 0-loss classifier on a sufficiently contaminated training set.*

*Proof.* We will proceed by contradiction.

Let patch $(x)$ denote the patched version of $x$. Without loss of generality, let the target label be $+1$. Set $\varepsilon_{\text{clean}}$ and $\varepsilon_{\text{adv}}$ such that $\varepsilon_{\text{clean}} + \varepsilon_{\text{adv}} < 1$ and draw enough samples such that the attack succeeds with parameters $\varepsilon_{\text{adv}}$ and $\delta$.

Observe that we can write every member in $S_{\text{adv}}$ as $(\text{patch}(x), y)$ for some natural $x$ with label $\neg y$. Next, suppose that the learner recovers a $\widehat{w}$ such that the empirical margin loss of $\widehat{w}$ is 0. Next, recall that the following holds for $\widehat{w}$ obtained from the minimization in the theorem statement and for a training set $S \sim \mathcal{D}^m$ (see, for instance, Theorem 26.12 of [5]):

$$\mathbb{E}_{(x,y) \sim \mathcal{D}}\left[\mathbb{1}\left\{y\left\langle\widehat{w}, x\right\rangle < 1\right\}\right] \leq \inf_{w \ : \ \|w\| \leq \gamma^{-1}} \mathbb{E}_{(x,y) \sim S}\left[\mathbb{1}\left\{y\left\langle w, x\right\rangle < 1\right\}\right] + O\left(\sqrt{\frac{\left(R/\gamma\right)^2 + \log\left(1/\delta\right)}{m}}\right)$$

Using this, it is easy to see that from uniform convergence, we have, with probability $1 - \delta$:

$$\Pr_{x \sim \mathcal{D}}\left[y\left\langle\widehat{w}, x\right\rangle \geq 1\right] \geq 1 - \varepsilon_{\text{clean}}$$

$$\Pr_{x \sim \mathcal{D}}\left[\left\langle\widehat{w}, \text{patch}(x)\right\rangle \geq 1\right] \geq 1 - \varepsilon_{\text{adv}}$$

Thus, by a Union Bound, the following must be true:

$$\Pr_{x \sim \mathcal{D}}\left[\left(y\left\langle\widehat{w}, x\right\rangle \geq 1\right) \wedge \left(\left\langle\widehat{w}, \text{patch}(x)\right\rangle \geq 1\right)\right] \geq 1 - \varepsilon_{\text{clean}} - \varepsilon_{\text{adv}}$$

Hence, it must be the case that there exists at least one true negative $x$ for which both $y\left\langle\widehat{w}, x\right\rangle \geq 1$ and $\left\langle\widehat{w}, \text{patch}(x)\right\rangle \geq 1$ hold. We will use this to obtain a lower bound on $\|\widehat{w}\|$, from which a contradiction will follow. Notice that:

$$\begin{aligned}
1 &\leq \left\langle\widehat{w}, \text{patch}(x)\right\rangle \\
&= \left\langle\widehat{w}, x\right\rangle + \left\langle\widehat{w}, \text{patch}(x) - x\right\rangle \\
&\leq -1 + \|\widehat{w}\| \cdot \|\text{patch}(x) - x\|
\end{aligned}$$

where the last line follows from the fact that $x$ is labeled differently from patch $(x)$. This gives:

$$\|\widehat{w}\| \geq \frac{2}{\|\text{patch}(x) - x\|}$$

Assuming that we meet the constraint $\|\widehat{w}\| \leq 1/\gamma$, putting the inequalities together gives:

$$\|\text{patch}(x) - x\| \geq 2\gamma$$

which is a contradiction, since we require that the size of the perturbation is smaller than the margin. $\qquad\square$

**Lemma 23** (Lemma 8). *We have $0 \leq \text{mcap}_{\mathcal{X}, \mathcal{D}}(\mathcal{H}) \leq \text{VC}(\mathcal{H})$.*

*Proof.* The lower bound is obvious. This is also tight, as we can set $\mathcal{X} = \{0,1\}^n$, $\mathcal{D} = \text{Unif}(\mathcal{X})$, and $\mathcal{H} = \{f \ : \ f(x) = 1, \forall x \in \mathcal{X}\}$.

We now tackle the upper bound. Suppose for the sake of contradiction that $\text{mcap}_{\mathcal{X}, \mathcal{D}}(\mathcal{H}) \geq \text{VC}(\mathcal{H}) + 1$. Then, we can find $k = \text{VC}(\mathcal{H}) + 1$ nonempty subsets of $\mathcal{X}$, $X_1, \ldots, X_k$ and an $h$ for which every labeling of these subsets can be achieved by some other $\widehat{h} \in \mathcal{H}$. Hence, picking any

collection of points $x_i \in X_i$ yields a set witnessing $\mathsf{VC}\left(\mathcal{H}\right) \geq k = \mathsf{VC}\left(\mathcal{H}\right) + 1$, which is clearly a contradiction.

The upper bound is tight as well. Consider the dataset $S = \{0, e_1, \ldots, e_d\}$, let $\mathcal{D}$ be a distribution assigning a point mass of 1 to $x = 0$, and let $h^*(0) = 1$. It is easy to see that the class of origin-containing halfspaces can memorize every labeling $e_1, \ldots, e_d$ as follows – suppose we have labels $b_1, \ldots, b_d$. Then, the classifier:

$$\mathbb{1}\left\{\sum_{i=1}^{d} b_i \cdot x_i \geq 0\right\}$$

memorizes every labeling of $e_1, \ldots, e_d$ while correctly classifying the pair $(0, 1)$. Hence, we can memorize $d$ irrelevant sets, which is equal to the VC dimension of origin-containing linear separators. $\qquad\square$

**Theorem 24** (Theorem 9). *Pick a target label $t \in \pm 1$. Suppose we have a hypothesis class $\mathcal{H}$, a target function $h^*$, a domain $\mathcal{X}$, a data distribution $\mathcal{D}$, and a class of patch functions $\mathcal{F}_{\mathsf{adv}}$. Define:*

$$\mathcal{C}(\mathcal{F}_{\mathsf{adv}}(h^*)) \coloneqq \{\mathsf{patch}\left(\mathsf{Supp}\left(\mathcal{D}|h^*(x) \neq t\right)\right) \ : \ \mathsf{patch} \in \mathcal{F}_{\mathsf{adv}}\}$$

*Now, suppose that $\mathsf{mcap}_{\mathcal{X},\mathcal{D}}\left(h^*, \mathcal{H}, \mathcal{C}(\mathcal{F}_{\mathsf{adv}}(h^*))\right) \geq 1$. Then, there exists a function $\mathsf{patch} \in \mathcal{F}_{\mathsf{adv}}$ for which the adversary can draw a set $S_{\mathsf{adv}}$ consisting of $m = \Omega\left(\varepsilon_{\mathsf{adv}}^{-1}\left(\mathsf{VC}\left(\mathcal{H}\right) + \log\left(1/\delta\right)\right)\right)$ i.i.d samples from $\mathcal{D}|h^*(x) \neq t$ such that with probability at least $1 - \delta$ over the draws of $S_{\mathsf{adv}}$, the adversary achieves the objectives of Problem 2, regardless of the number of samples the learner draws from $\mathcal{D}$ for $S_{\mathsf{clean}}$.*

**Theorem 25** (Generalization of Theorem 9). *Pick an array of $k$ target labels $t \in \{\pm 1\}^k$. Suppose we have a hypothesis class $\mathcal{H}$, a target function $h^*$, a domain $\mathcal{X}$, a data distribution $\mathcal{D}$, and a class of patch functions $\mathcal{F}_{\mathsf{adv}}$. Define:*

$$\mathcal{C}(\mathcal{F}_{\mathsf{adv}}(h^*))_{t'} \coloneqq \{\mathsf{patch}\left(\mathsf{Supp}\left(\mathcal{D}|h^*(x) \neq t'\right)\right) \ : \ \mathsf{patch} \in \mathcal{F}_{\mathsf{adv}}\}$$

*and let:*

$$\mathcal{C}(\mathcal{F}_{\mathsf{adv}}(h^*)) \coloneqq \mathcal{C}(\mathcal{F}_{\mathsf{adv}}(h^*))_{-1} \cup \mathcal{C}(\mathcal{F}_{\mathsf{adv}}(h^*))_1$$

*Now, suppose that $\mathsf{mcap}_{\mathcal{X},\mathcal{D}}\left(h^*, \mathcal{H}, \mathcal{C}(\mathcal{F}_{\mathsf{adv}}(h^*))\right) \geq k$. Then, there exists $k$ functions $\mathsf{patch}_1, \ldots, \mathsf{patch}_k \in \mathcal{F}_{\mathsf{adv}}$ for which the adversary can draw sets $\{(S_{\mathsf{adv}})_i\}_{i \in [k]}$ each consisting of $m_i = \Omega\left(\varepsilon_{\mathsf{adv}}^{-1}\left(\mathsf{VC}\left(\mathcal{H}\right) + \log\left(k/\delta\right)\right)\right)$ i.i.d samples from $\mathcal{D}|h^*(x) \neq t_i$ such that with probability at least $1 - \delta$ over the draws of $(S_{\mathsf{adv}})_i$, the adversary achieves the objectives of Problem 2, regardless of the number of samples the learner draws from $\mathcal{D}$ for $S_{\mathsf{clean}}$.*

*Proof.* As per the theorem statement, we can draw $m$ samples from $\mathcal{D}|h^*(x) \neq t_i$ to form $S_{\mathsf{adv}}$ by inverting the labels of the samples we draw.

Since $\mathsf{mcap}_{\mathcal{X},\mathcal{D}}\left(h^*, \mathcal{H}, \mathcal{C}(\mathcal{F}_{\mathsf{adv}}(h^*))\right) = k$, there must exist $k$ sets $X_1, \ldots, X_k \in \mathcal{C}(\mathcal{F}_{\mathsf{adv}}(h^*))$ such that the $X_i$ are memorizable, for which we can write $X_i \subseteq \mathsf{patch}_i\left(\mathsf{Supp}\left(\mathcal{D}|h^*(x) \neq t_i\right)\right)$ for appropriate choices of $\mathsf{patch}_i$, and for which $\mu_{\mathsf{patch}(\mathcal{D}|h^*(x) \neq t_i)}(X_i) = 1$. This implies that with probability 1, there exists at least one function $\widehat{h} \in \mathcal{H}$ such that $\widehat{h}$ returns $t_i$ on every element in $(S_{\mathsf{adv}})_i$ for all $i \in [k]$ and agrees with $h^*$ on every element in the clean training set $S_{\mathsf{clean}}$.

Thus, we can recover a classifier $\widehat{h}$ from $\mathcal{H}$ with 0 error on the training set $S_{\mathsf{clean}} \cup \left(\bigcup_{i \in [k]}(S_{\mathsf{adv}})_i\right)$. In particular, notice that we achieve 0 error on $S_{\mathsf{clean}}$ from distribution $\mathcal{D}$ and on every $(S_{\mathsf{adv}})_i$ from distribution $\mathsf{patch}_i\left(\mathcal{D}|h^*(x) \neq t_i\right)$. From the Fundamental Theorem of PAC Learning ([5]), it follows that as long as $|S_{\mathsf{clean}}|$ and $|(S_{\mathsf{adv}})_i|$ are each at least $\Omega\left(\varepsilon_{\mathsf{clean}}^{-1}\left(\mathsf{VC}\left(\mathcal{H}\right) + \log\left(k/\delta\right)\right)\right)$ and $\Omega\left(\varepsilon_{\mathsf{adv}}^{-1}\left(\mathsf{VC}\left(\mathcal{H}\right) + \log\left(k/\delta\right)\right)\right)$, respectively, we have that $\widehat{h}$ has error at most $\varepsilon$ on $\mathcal{D}$ and error at least $1 - \varepsilon$ on $\mathsf{patch}_i\left(\mathcal{D}|h^*(x) \neq t_i\right)$ with probability $1 - \delta$ (following from a union bound, where each training subset yields a failure to attain uniform convergence with probability at most $\delta/(k+1)$). $\qquad\square$

**Theorem 26** (Theorem 10). *Let $\mathcal{C}(\mathcal{F}_{\mathsf{adv}}(h^*))$ be defined the same as in Theorem 9. Suppose we have a hypothesis class $\mathcal{H}$ over a domain $\mathcal{X}$, a true classifier $h^*$, data distribution $\mathcal{D}$, and a perturbation class $\mathcal{F}_{\mathsf{adv}}$. If $\mathsf{mcap}_{\mathcal{X},\mathcal{D}}\left(h^*, \mathcal{H}, \mathcal{C}(\mathcal{F}_{\mathsf{adv}}(h^*))\right) = 0$, then the adversary cannot successfully construct a backdoor data poisoning attack as per the conditions of Problem 2.*

*Proof.* The condition in the theorem statement implies that there does not exist an irrelevant set that can be memorized atop any choice of $h \in \mathcal{H}$.

For the sake of contradiction, suppose that there does exist a target classifier $h^*$, a function patch $\in \mathcal{F}_{\text{adv}}$ and a target label $t$ such that for all choices of $\varepsilon_{\text{clean}}$, $\varepsilon_{\text{adv}}$, and $\delta$, we obtain a successful attack.

Define the set $X := \text{patch}\left(\text{Supp}\left(\mathcal{D}|h^*(x) \neq t\right)\right)$; in words, $X$ is the subset of $\mathcal{X}$ consisting of patched examples that are originally of the opposite class of the the target label. It is easy to see that $X \in \mathcal{C}$.

We will first show that if $\mu_{\mathcal{D}}(X) > 0$, then we obtain a contradiction. Set $0 < \varepsilon_{\text{adv}}, \varepsilon_{\text{clean}} < \frac{\mu_{\mathcal{D}}(X)}{1+\mu_{\mathcal{D}}(X)}$. Since the attack is successful, we must classify at least a $1 - \varepsilon_{\text{adv}}$ fraction of $X$ as the target label. Hence, we can write:

$$\mu_{\mathcal{D}}\left(\left\{x \in X \;:\; \widehat{h}(x) = t\right\}\right) \geq \left(1 - \varepsilon_{\text{adv}}\right)\mu_{\mathcal{D}}(X)$$

$$> \frac{1}{1 + \mu_{\mathcal{D}}(X)} \cdot \mu_{\mathcal{D}}(X)$$

$$> \varepsilon_{\text{clean}}$$

Since the set $\left\{x \in X \;:\; \widehat{h}(x) = t\right\}$ is a subset of the region of $\mathcal{X}$ that $\widehat{h}$ makes a mistake on, we have that $\widehat{h}$ must make a mistake on at least $\varepsilon_{\text{clean}}$ measure of $\mathcal{D}$, which is a contradiction.

Hence, it must be the case that $\mu_{\mathcal{D}}(X) = 0$; in other words, $X$ is an irrelevant set. Recall that in the beginning of the proof, we assume there exists a function $\widehat{h}$ that achieves label $t$ on $X$, which is opposite of the value of $h^*$ on $X$. Since we can achieve both possible labelings of $X$ with functions from $\mathcal{H}$, it follows that $X$ is a memorizable set, and thus the set $X$ witnesses $\text{mcap}_{\mathcal{X},\mathcal{D}}\left(h^*, \mathcal{H}, \mathcal{C}(\mathcal{F}_{\text{adv}}(h^*))\right)$. $\qquad\square$

**Example 27** (Overparameterized Linear Classifiers (Example 11))**.** *Recall the result from the previous section, where we took $\mathcal{X} = \mathbb{R}^d$, $\mathcal{H}_d$ to be the set of linear classifiers in $\mathbb{R}^d$, and let $\mathcal{D}$ be a distribution over a radius-$R$ subset of an $s$-dimensional subspace $P$. We also assume that the true labeler $h^*$ achieves margin $\gamma$.*

*If we set $\mathcal{F}_{\text{adv}} = \left\{\text{patch}(x) \;:\; \text{patch}(x) = x + \eta, \eta \in \mathbb{R}^d\right\}$, then we have $\text{mcap}_{\mathcal{X},\mathcal{D}}\left(h^*, \mathcal{H}_d, \mathcal{C}(\mathcal{F}_{\text{adv}}(h^*))\right) \geq d - s$.*

*Proof.* Let $w^*$ be the weight vector corresponding to $h^*$.

Observe that there exists $k := d - s$ unit vectors $v_1, \ldots, v_k$ that complete an orthonormal basis from that for $P$ to one for $\mathbb{R}^d$. Next, consider the following subset of $\mathcal{F}_{\text{adv}}(h^*)$:

$$\mathcal{F}'_{\text{adv}} := \left\{\text{patch} \in \mathcal{F}_{\text{adv}} \;:\; \forall i \in [k], \text{patch}_i(x) = \left(\begin{cases} x + \eta \cdot t_i v_i & , h^*(x) \neq t_i \\ x & \text{otherwise} \end{cases}\right)\right\}$$

We prove the memorization capacity result by using the images of functions in $\mathcal{F}'_{\text{adv}}$. We will show that the function:

$$\widehat{h}(x) = \text{sign}\left(\left\langle w^* + \frac{2R}{\gamma}\sum_{i=1}^{k} t_i \cdot \frac{v_i}{\eta_i}, x \right\rangle\right)$$

memorizes the $k$ sets $C_i := \left\{x + \eta_i \cdot v_i \;:\; \langle w^*, x \rangle \in [1, R/\gamma] \cup [-R/\gamma, -1]\right\}$. Moreover, observe that the preimages of the $C_i$ have measure 1 under the conditional distributions $\mathcal{D}|h^*(x) \neq t_i$, since the preimages contain the support of these conditional distributions. We now have that, for a clean point $x \in P$:

$$\widehat{h}(x) = \text{sign}\left(\left\langle w^* + \frac{2R}{\gamma}\sum_{i=1}^{k} t_i \cdot \frac{v_i}{\eta_i}, x \right\rangle\right)$$

$$= \text{sign}\left(\langle w^*, x \rangle + \frac{2R}{\gamma}\left\langle \sum_{i=1}^{k} t_i \cdot \frac{v_i}{\eta_i}, x \right\rangle\right)$$

$$= \text{sign}\left(\langle w^*, x \rangle\right) = h^*(x)$$

and for a corrupted point $x + \eta_j \cdot v_j$, for $j \in [k]$:

$$\widehat{h}(x) = \text{sign}\left(\left\langle w^* + \frac{2R}{\gamma}\sum_{i=1}^{k} t_i \cdot \frac{v_i}{\eta_i}, x + \eta_j \cdot v_j \right\rangle\right)$$

$$= \text{sign}\left(\langle w^*, x + \eta_j \cdot v_j \rangle + \frac{2R}{\gamma}\left\langle \sum_{i=1}^{k} t_i \cdot \frac{v_j}{\eta_j}, x + \eta_j \cdot v_j \right\rangle\right)$$

$$= \text{sign}\left(\langle w^*, x \rangle + \frac{2R}{\gamma}\left\langle \sum_{i=1}^{k} t_i \cdot \frac{v_i}{\eta_i}, x \right\rangle + \frac{2R}{\gamma}\left\langle \sum_{i=1}^{k} t_i \cdot \frac{v_i}{\eta_i}, \eta_j \cdot v_j \right\rangle\right)$$

$$= \text{sign}\left(\left[\pm\frac{R}{\gamma}\right] + t_j \cdot \frac{2R}{\gamma}\right)$$

$$= t_j$$

This shows that we can memorize the $k$ sets $C_i$. It is easy to see that $\mu_{\mathcal{D}}(C_i) = 0$, so the $C_i$ are irrelevant memorizable sets; in turn, we have that $\text{mcap}_{\mathcal{X},\mathcal{D}}(h^*) \geq k = d - s$, as desired. $\qquad\square$

**Example 28** (Linear Classifiers Over Convex Bodies (Example 12)). *Let $\mathcal{H}$ be the set of origin-containing halfspaces. Fix an origin-containing halfspace $h^*$ with weight vector $w^*$. Let $\mathcal{X}'$ be a closed compact convex set, let $\mathcal{X} = \mathcal{X}' \setminus \{x : \langle w^*, x \rangle = 0\}$, and let $\mathcal{D}$ be any probability measure over $\mathcal{X}$ that assigns nonzero measure to every $\ell_2$ ball of nonzero radius contained in $\mathcal{X}$ and satisfies the relation $\mu_{\mathcal{D}}(Y) = 0 \iff \text{Vol}_d(Y) = 0$ for all $Y \subset \mathcal{X}$. Then, $\text{mcap}_{\mathcal{X},\mathcal{D}}(h^*, \mathcal{H}) = 0$.*

*Proof.* Observe that it must be the case that the dimension of the ambient space is equal to the dimension of $\mathcal{X}$.

Let $w^*$ be the weight vector corresponding to the true labeler $h^*$.

For the sake of contradiction, suppose there exists a classifier $\widehat{w}$ satisfying $\Pr_{x \sim \mathcal{D}}\left[\text{sign}\left(\langle \widehat{w}, x \rangle\right) = \text{sign}\left(\langle w^*, x \rangle\right)\right] = 1$, but there exists a subset $Y \subset \mathcal{X}$ for which $\text{sign}\left(\langle \widehat{w}, x \rangle\right) \neq \text{sign}\left(\langle w^*, x \rangle\right)$, for all $x \in Y$. Such a $Y$ would constitute a memorizable set.

Without loss of generality, let the target label be $-1$; that is, the adversary is converting a set $Y$ whose label is originally $+1$ to one whose label is $-1$. Additionally, without loss of generality, take $\|w^*\| = \|\widehat{w}\| = 1$. Observe that the following set relationship must hold:

$$Y \subseteq D := \{x \in \mathcal{X} : \langle \widehat{w}, x \rangle \leq 0 \text{ and } \langle w^*, x \rangle > 0\}$$

For $D$ to be nonempty (and therefore for $Y$ to be nonempty), observe that we require $\widehat{w} \neq w^*$ (otherwise, the constraints in the definition of the set $D$ are unsatisfiable).

We now need the following intermediate result.

**Lemma.** *Consider some convex body $K$, a probability measure $\mathcal{D}$ such that every $\ell_2$ ball of nonzero radius within $K$ has nonzero measure, and some subset $K' \subseteq K$ satisfying $\mu_{\mathcal{D}}(K') = 1$. Then, $\text{conv}(K')$ contains every interior point of $K$.*

*Proof.* Recall that an interior point is defined as one for which we can find some neighborhood contained entirely within the convex body. Mathematically, $x \in K$ is an interior point if we can find nonzero $\delta$ for which $\{z : \|x - z\| \leq \delta\} \subseteq K$ (see [1]).

For the sake of contradiction, suppose that there exists some interior point $x \in K$ that is not contained in $\text{conv}(K')$. Hence, there must exist a halfspace $H$ with boundary passing through $x$ and entirely containing $\text{conv}(K')$. Furthermore, there must exist a nonzero $\delta$ for which there is an $\ell_2$ ball centered at $x$ of radius $\delta$ contained entirely within $K$. Call this ball $B_2(x, \delta)$. Thus, the set $K \setminus H$ cannot be in $\text{conv}(K')$.

We will now show that $\mu_{\mathcal{D}}(K \setminus H) > 0$. Observe that the hyperplane inducing $H$ must cut $B_2(x, \delta)$ through an equator. From this, we have that the set $K \setminus H$ contains a half-$\ell_2$ ball of radius $\delta$. It is

easy to see that this half-ball contains another $\ell_2$ ball of radius $\delta/2$ (call this $B'$), and as per our initial assumption, $B'$ must have nonzero measure.

Thus, we can write $\mu_{\mathcal{D}}(K \backslash H) \geq \mu_{\mathcal{D}}(B') > 0$. Since we know that $\mu_{\mathcal{D}}(\text{conv}\,(K')) + \mu_{\mathcal{D}}(K \backslash H) \leq 1$, it follows that $\mu_{\mathcal{D}}(\text{conv}\,(K')) < 1$ and therefore $\mu_{\mathcal{D}}(K') < 1$, violating our initial assumption that $\mu_{\mathcal{D}}(K') = 1$. $\qquad \square$

This lemma implies that if $Y$ is memorizable, then it must lie entirely on the boundary of the set $\mathcal{X}_+ := \{x \in \mathcal{X} \; : \; \langle w^*, x \rangle > 0\}$. To see this, observe that if $\widehat{w}$ classifies any (conditional) measure-1 subset of $\mathcal{X}_+$ correctly, then it must classify the convex hull of that subset correctly as well. This implies that $\widehat{w}$ must correctly classify every interior point in $\mathcal{X}_+$, and thus, $Y$ must be entirely on the boundary of $\mathcal{X}_+$.

We will now show the following intermediate result.

**Lemma.** *Let $K$ be a closed compact convex set. Let $x_1$ be on the boundary of $K$ and let $x_2$ be an interior point of $K$. Then, every point of the form $\lambda x_1 + (1 - \lambda)x_2$ for $\lambda \in (0, 1)$ is an interior point of $K$.*

*Proof.* Since $x_2$ is an interior point, there must exist an $\ell_2$ ball of radius $\delta$ contained entirely within $K$ centered at $x_2$. From similar triangles and the fact that any two points in a convex body can be connected by a line contained in the convex body, it is easy to see that we can center an $\ell_2$ ball of radius $(1 - \lambda)\delta$ at the point $\lambda x_1 + (1 - \lambda)x_2$ that lies entirely in $K$. This is what we wanted, and we're done. $\qquad \square$

Now, let $x_1 \in Y$ and $x_2 \in \text{Interior}(\mathcal{X}_-)$ where $\mathcal{X}_- = \{x \in \mathcal{X} \; : \; \langle w^*, x \rangle < 0\}$. Draw a line from $x_1$ to $x_2$ and consider the labels of the points assigned by $\widehat{w}$. Since $x_1 \in Y$, we have $\widehat{h}(x_1) = -1$, and since $x_2 \in \text{Interior}(\mathcal{X}_-)$, we have that $\widehat{h}(x_2) = -1$ as well. Using our lemma, we have that every point on the line connecting $x_1$ to $x_2$ (except for possibly $x_1$) is an interior point to $\mathcal{X}'$. Since we have that the number of sign changes along a line that can be induced by a linear classifier is at most 1, we must have that the line connecting $x_1$ to $x_2$ incurs 0 sign changes with respect to the classifier induced by $\widehat{w}$. This implies that the line connecting $x_1$ to $x_2$ cannot pass through any interior points of $\mathcal{X}_+$. However, the only way that this can happen is if $\langle w^*, x_1 \rangle = 0$, but per our definition of $\mathcal{X}$, if it is the case that $\langle w^*, x_1 \rangle = 0$, then $x_1 \notin \mathcal{X}$, which is a clear contradiction.

This is sufficient to conclude the proof, and we're done. $\qquad \square$

**Example 29** (Sign Changes (Example 13)). *Let $\mathcal{X} = [0, 1]$, $\mathcal{D} = \text{Unif}\,(\mathcal{X})$ and $\mathcal{H}_k$ be the class of functions admitting at most $k$ sign-changes. Specifically, $\mathcal{H}_k$ consists of functions $h$ for which we can find pairwise disjoint, continuous intervals $I_1, \ldots, I_{k+1}$ such that:*

- *For all $i < j$ and for all $x \in I_i, y \in I_j$, we have $x < y$.*
- *$\bigcup_{i=1}^{k+1} I_i = \mathcal{X}$.*
- *$h(I_i) = -h(I_{i+1})$, for all $i \in [k]$.*

*Suppose the learner is learning $\mathcal{H}_s$ for unknown $s$ using $\mathcal{H}_d$, where $s \leq d + 2$. For all $h^* \in \mathcal{H}_s$, we have $\text{mcap}_{\mathcal{X}, \mathcal{D}}\,(h^*, \mathcal{H}_d) \geq \lfloor (d-s)/2 \rfloor$.*

*Proof.* Without loss of generality, take $d - s$ to be an even integer.

Let $I_1, \ldots, I_{s+1}$ be the intervals associated with $h^*$. It is easy to see that we can pick a total of $(d-s)/2$ points such that the sign of these points can be memorized by some $\widehat{h}$. Since each point we pick within an interval can induce at most 2 additional sign changes, we have that the resulting function $\widehat{h}$ has at most $s + 2 \cdot (d-s)/2 \leq d$ sign-changes; thus, $\widehat{h} \in \mathcal{H}_d$. Moreover, the measure of a single point is 0, and so the total measure of our $(d-s)/2$ points is 0.

Given this, it is easy to find $\mathcal{F}_{\text{adv}}$ and corresponding $\mathcal{C}(\mathcal{F}_{\text{adv}}(h^*))$ for which the backdoor attack can succeed as per Theorem 9. $\qquad \square$

**Theorem 30** (Theorem 14). *Suppose that the learner can calculate and minimize:*

$$\mathcal{L}_{\mathcal{F}_{\mathsf{adv}}(h^*)}(\widehat{h}, S) = \mathop{\mathbb{E}}_{(x,y)\sim S}\left[\sup_{\mathsf{patch}\in\mathcal{F}_{\mathsf{adv}}(h^*)} \mathbb{1}\left\{\widehat{h}(\mathsf{patch}\,(x)) \neq y\right\}\right]$$

*over a finite set $S$ and $\widehat{h} \in \mathcal{H}$.*

*If the VC dimension of the loss class $\mathcal{L}^{\mathcal{H}}_{\mathcal{F}_{\mathsf{adv}}(h^*)}$ is finite, then there exists an algorithm using $O\left(\varepsilon_{\mathsf{clean}}^{-2}\left(\mathsf{VC}\left(\mathcal{L}_{\mathcal{F}_{\mathsf{adv}}(h^*)}\right) + \log\left(1/\delta\right)\right)\right)$ samples that allows the learner to defeat the adversary through learning a backdoor-robust classifier or by rejecting the training set as being corrupted, with probability $1 - \delta$.*

*Proof.* See Algorithm A.1 for the pseudocode of an algorithm witnessing Theorem 17.

---

**Algorithm A.1** Implementation of an algorithm certifying backdoor corruption

1: **Input**: Training set $S = S_{\mathsf{clean}} \cup S_{\mathsf{adv}}$
   satisfying $|S_{\mathsf{clean}}| = \Omega\left(\varepsilon_{\mathsf{clean}}^{-2}\left(\mathsf{VC}\left(\mathcal{L}^{\mathcal{H}}_{\mathcal{F}_{\mathsf{adv}}(h^*)}\right) + \log\left(1/\delta\right)\right)\right)$
2: Set $\widehat{h} := \mathrm{argmin}_{h\in\mathcal{H}}\mathcal{L}_{\mathcal{F}_{\mathsf{adv}}(h^*)}(h, S)$
3: **Output**: $\widehat{h}$ if $\mathcal{L}_{\mathcal{F}_{\mathsf{adv}}(h^*)}(\widehat{h}, S) \leq 2\varepsilon$ and reject otherwise

---

There are two scenarios to consider.

**Training set is (mostly) clean.** Suppose that $S$ satisfies $\min_{h\in\mathcal{H}}\mathcal{L}_{\mathcal{F}_{\mathsf{adv}}(h^*)}(h, S) \lesssim \varepsilon_{\mathsf{clean}}$. Since the VC dimension of the loss class $\mathcal{L}^{\mathcal{H}}_{\mathcal{F}_{\mathsf{adv}}(h^*)}$ is finite, it follows that with finitely many samples, we attain uniform convergence with respect to the robust loss, and we're done; in particular, we can write $\mathcal{L}_{\mathcal{F}_{\mathsf{adv}}(h^*)}\left(\mathrm{argmin}_{h\in\mathcal{H}}\mathcal{L}_{\mathcal{F}_{\mathsf{adv}}(h^*)}(h, S), \mathcal{D}\right) \lesssim \varepsilon_{\mathsf{clean}}$ with high probability.

**Training set contains many backdoored examples.** Here, we will show that with high probability, minimizing $\mathcal{L}_{\mathcal{F}_{\mathsf{adv}}(h^*)}(\widehat{h}, S)$ over $\widehat{h}$ will result in a nonzero loss, which certifies that the training set $S$ consists of malicious examples.

Suppose that for the sake of contradiction, the learner finds a classifier $\widehat{h}$ such that $\mathcal{L}_{\mathcal{F}_{\mathsf{adv}}(h^*)}(\widehat{h}, S) \lesssim \varepsilon_{\mathsf{clean}}$. Hence, with high probability, we satisfy $\mathcal{L}_{\mathcal{F}_{\mathsf{adv}}(h^*)}(\widehat{h}, \mathcal{D}) \lesssim \varepsilon_{\mathsf{clean}}$. Since there is a constant measure allocated to each class, we can write:

$$\mathop{\mathbb{E}}_{(x,y)\sim\mathcal{D}|y\neq t}\left[\sup_{\mathsf{patch}\in\mathcal{F}_{\mathsf{adv}}(h^*)} \mathbb{1}\left\{\widehat{h}(\mathsf{patch}\,(x)) \neq y\right\}\right] \lesssim \varepsilon_{\mathsf{clean}}$$

Furthermore, since we achieved a loss of $0$ on the whole training set, including the subset $S_{\mathsf{adv}}$, from uniform convergence, we satisfy the following with high probability:

$$\mathop{\mathbb{E}}_{(x,y)\sim\mathcal{D}|y\neq t}\left[\mathbb{1}\left\{\widehat{h}(\mathsf{patch}\,(x)) = t\right\}\right] \geq 1 - \varepsilon_{\mathsf{adv}}$$

which immediately implies:

$$\mathop{\mathbb{E}}_{(x,y)\sim\mathcal{D}|y\neq t}\left[\sup_{\mathsf{patch}\in\mathcal{F}_{\mathsf{adv}}(h^*)} \mathbb{1}\left\{\widehat{h}(\mathsf{patch}\,(x)) \neq y\right\}\right] \geq 1 - \varepsilon_{\mathsf{adv}}$$

Chaining the inequalities together yields:

$$\varepsilon_{\mathsf{clean}} \gtrsim 1 - \varepsilon_{\mathsf{adv}}$$

which is a contradiction, as we can make $\varepsilon_{\mathsf{clean}}$ sufficiently small so as to violate this statement. $\square$

**Theorem 31** (Filtering Implies Generalization (Theorem 17)). *Let $\alpha \leq 1/3$ and $\varepsilon_{\mathsf{clean}} \leq 1/10$.*

*Suppose we have a training set $S = S_{\mathsf{clean}} \cup S_{\mathsf{adv}}$ such that $|S_{\mathsf{clean}}| = \Omega\left(\varepsilon_{\mathsf{clean}}^{-2}\left(\mathsf{VC}\left(\mathcal{L}_{\mathcal{F}_{\mathsf{adv}}(h^*)}\right) + \log\left(1/\delta\right)\right)\right)$ and $|S_{\mathsf{adv}}| \leq \alpha \cdot \left(|S_{\mathsf{adv}}| + |S_{\mathsf{clean}}|\right)$. If there exists an algorithm that given $S$ can find a subset $S' = S'_{\mathsf{clean}} \cup S'_{\mathsf{adv}}$ satisfying $|S'_{\mathsf{clean}}|/|S_{\mathsf{clean}}| \geq 1 - \varepsilon_{\mathsf{clean}}$ and $\min_{h\in\mathcal{H}}\mathcal{L}_{\mathcal{F}_{\mathsf{adv}}(h^*)}(h, S') \lesssim \varepsilon_{\mathsf{clean}}$, then there exists an algorithm such that given $S$ returns a function $\widehat{h}$ satisfying $\mathcal{L}_{\mathcal{F}_{\mathsf{adv}}(h^*)}(\widehat{h}, \mathcal{D}) \lesssim \varepsilon_{\mathsf{clean}}$ with probability $1 - \delta$.*

*Proof.* See Algorithm A.2 for the pseudocode of an algorithm witnessing the theorem statement.

---

**Algorithm A.2** Implementation of a generalization algorithm given an implementation of a filtering algorithm

---

1: **Input**: Training set $S = S_{\text{clean}} \cup S_{\text{adv}}$
   satisfying $|S_{\text{clean}}| = \Omega\left(\varepsilon_{\text{clean}}^{-2}\left(\text{VC}\left(\mathcal{L}_{\mathcal{F}_{\text{adv}}(h^*)}\right) + \log\left(1/\delta\right)\right)\right)$
2: Run the filtering algorithm on $S$ to obtain $S'$ satisfying the conditions in the theorem statement
3: **Output**: Output $\hat{h}$, defined as $\hat{h} := \operatorname{argmin}_{h \in \mathcal{H}} \mathcal{L}_{\mathcal{F}_{\text{adv}}(h^*)}(h, S')$

---

Recall that we have drawn enough samples to achieve uniform convergence (see [14] and [26]); in particular, assuming that our previous steps succeeded in removing very few points from $S_{\text{clean}}$, then for all $h \in \mathcal{H}$, we have with probability $1 - \delta$:

$$\left|\mathcal{L}_{\mathcal{F}_{\text{adv}}(h^*)}(h, \mathcal{D}) - \mathcal{L}_{\mathcal{F}_{\text{adv}}(h^*)}(h, S_{\text{clean}})\right| \leq \varepsilon_{\text{clean}}$$

Observe that we have deleted at most $m \cdot 2\varepsilon_{\text{clean}}$ points from $S_{\text{clean}}$. Let $S'_{\text{clean}} := S' \cap S_{\text{clean}}$ (i.e., the surviving members of $S_{\text{clean}}$ from our filtering procedure). We start with the following claim.

**Claim 32.** *The following holds for all $h \in \mathcal{H}$:*

$$\left|\mathcal{L}_{\mathcal{F}_{\text{adv}}(h^*)}(h, S_{\text{clean}}) - \mathcal{L}_{\mathcal{F}_{\text{adv}}(h^*)}(h, S'_{\text{clean}})\right| \leq \varepsilon_{\text{clean}}$$

*Proof.* Let $a, b, c$ be positive numbers. We first write:

$$\frac{a}{b} - \max\left\{0, \frac{a-c}{b-c}\right\} = \frac{c(b-a)}{b(b-c)} \leq \frac{c}{b}$$

which occurs exactly when $c \leq a$. In case where $a \leq c$:

$$\frac{a}{b} - \max\left\{0, \frac{a-c}{b-c}\right\} = \frac{a}{b} \leq \frac{c}{b}$$

which gives:

$$\frac{a}{b} - \max\left\{0, \frac{a-c}{b-c}\right\} \leq \frac{c}{b}$$

Next, consider:

$$\min\left\{1, \frac{a}{b-c}\right\} - \frac{a}{b} = \frac{a}{b-c} - \frac{a}{b} = \frac{c}{b} \cdot \frac{a}{b-c} \leq \frac{c}{b}$$

which happens exactly when we have $b \geq a + c$. In the other case:

$$\min\left\{1, \frac{a}{b-c}\right\} - \frac{a}{b} = 1 - \frac{a}{b} \leq \frac{c}{b}$$

We can thus write:

$$\max\left\{0, \frac{a-c}{b-c}\right\}, \min\left\{1, \frac{a}{b-c}\right\} \in \left[\frac{a}{b} \pm \frac{c}{b}\right]$$

Now, let $a$ denote the number of samples from $S_{\text{clean}}$ that $h$ incurs robust loss on, let $b$ be the total number of samples from $S_{\text{clean}}$, and let $c$ be the number of samples our filtering procedure deletes from $S_{\text{clean}}$. It is easy to see that $a/b$ corresponds $\mathcal{L}_{\mathcal{F}_{\text{adv}}(h^*)}(h, S_{\text{clean}})$ and that $\mathcal{L}_{\mathcal{F}_{\text{adv}}(h^*)}(h, S'_{\text{clean}}) \in [\max\{0, (a-c)/(b-c)\}, \min\{1, a/(b-c)\}]$. From our argument above, this means that we must have:

$$\mathcal{L}_{\mathcal{F}_{\text{adv}}(h^*)}(h, S'_{\text{clean}}) \in \left[\mathcal{L}_{\mathcal{F}_{\text{adv}}(h^*)}(h, S_{\text{clean}}) \pm \frac{\varepsilon_{\text{clean}}(1-\alpha)m}{(1-\alpha)m}\right]$$

Finally:

$$\frac{\varepsilon_{\text{clean}}(1-\alpha)m}{(1-\alpha)m} = \varepsilon_{\text{clean}}$$

and we're done. □

We now use our claim and triangle inequality to write:

$$\left| \mathcal{L}_{\mathcal{F}_{\text{adv}}(h^*)}(h, S'_{\text{clean}}) - \mathcal{L}_{\mathcal{F}_{\text{adv}}(h^*)}(h, \mathcal{D}) \right| \leq \left| \mathcal{L}_{\mathcal{F}_{\text{adv}}(h^*)}(h, S_{\text{clean}}) - \mathcal{L}_{\mathcal{F}_{\text{adv}}(h^*)}(h, S'_{\text{clean}}) \right| +$$
$$\left| \mathcal{L}_{\mathcal{F}_{\text{adv}}(h^*)}(h, \mathcal{D}) - \mathcal{L}_{\mathcal{F}_{\text{adv}}(h^*)}(h, S_{\text{clean}}) \right|$$
$$\leq \varepsilon_{\text{clean}}$$

Next, consider some $\widehat{h}$ satisfying $\mathcal{L}_{\mathcal{F}_{\text{adv}}(h^*)}(\widehat{h}, S') \lesssim \varepsilon_{\text{clean}}$ (which must exist, as per our argument in Part 3), and observe that, for a constant $C$:

$$\mathcal{L}_{\mathcal{F}_{\text{adv}}(h^*)}(\widehat{h}, S') \geq (1 - C\varepsilon_{\text{clean})}\mathcal{L}_{\mathcal{F}_{\text{adv}}(h^*)}(\widehat{h}, S' \cap S_{\text{clean}}) + C\varepsilon_{\text{clean}}\mathcal{L}_{\mathcal{F}_{\text{adv}}(h^*)}(\widehat{h}, S' \cap S_{\text{adv}})$$
$$\geq (1 - C\varepsilon_{\text{clean}})\mathcal{L}_{\mathcal{F}_{\text{adv}}(h^*)}(\widehat{h}, S'_{\text{clean}})$$
$$\Rightarrow \mathcal{L}_{\mathcal{F}_{\text{adv}}(h^*)}(\widehat{h}, S'_{\text{clean}}) \leq \frac{\varepsilon_{\text{clean}}}{1 - C\varepsilon_{\text{clean}}} = 2\varepsilon_{\text{clean}}\left(\frac{1}{1 - C\varepsilon_{\text{clean}}}\right) \lesssim \varepsilon_{\text{clean}}$$

We now use the fact that $\left| \mathcal{L}_{\mathcal{F}_{\text{adv}}(h^*)}(h, S'_{\text{clean}}) - \mathcal{L}_{\mathcal{F}_{\text{adv}}(h^*)}(h, \mathcal{D}) \right| \leq \varepsilon_{\text{clean}}$ to arrive at the conclusion that $\mathcal{L}_{\mathcal{F}_{\text{adv}}(h^*)}(h, \mathcal{D}) \lesssim \varepsilon_{\text{clean}}$, which is what we wanted to show. $\qquad\square$

**Theorem 33** (Generalization Implies Filtering (Theorem 18)). *Set $\varepsilon_{\text{clean}} \leq 1/10$ and $\alpha \leq 1/6$.*

*If there exists an algorithm that, given at most a $2\alpha$ fraction of outliers in the training set, can output a hypothesis satisfying $\mathcal{L}_{\mathcal{F}_{\text{adv}}(h^*)}(\widehat{h}, \mathcal{D}) \leq \varepsilon_{\text{clean}}$ with probability $1 - \delta$ over the draw of the training set, then there exists an algorithm that given a training set $S = S_{\text{clean}} \cup S_{\text{adv}}$ satisfying $|S_{\text{clean}}| \geq \Omega\left(\varepsilon_{\text{clean}}^{-2}\left(\text{VC}\left(\mathcal{L}_{\mathcal{F}_{\text{adv}}(h^*)}\right) + \log\left(1/\delta\right)\right)\right)$ outputs a subset $S' \subseteq S$ with the property that $\mathcal{L}_{\mathcal{F}_{\text{adv}}(h^*)}\left(\text{argmin}_{h \in \mathcal{H}}\mathcal{L}_{\mathcal{F}_{\text{adv}}(h^*)}(h, S'), \mathcal{D}\right) \lesssim \varepsilon_{\text{clean}}$ with probability $1 - 7\delta$.*

*Proof.* See Algorithm A.3 for the pseudocode of an algorithm witnessing the theorem statement.

At a high level, our proof proceeds as follows. We first show that the partitioning step results in partitions that don't have too high of a fraction of outliers, which will allow us to call the filtering procedure without exceeding the outlier tolerance. Then, we will show that the hypotheses $\widehat{h}_L$ and $\widehat{h}_R$ mark most of the backdoor points for deletion while marking only few of the clean points for deletion. Finally, we will show that although $\widehat{h}$ is learned on $S'$ that is not sampled i.i.d from $\mathcal{D}$, $\widehat{h}$ still generalizes to $\mathcal{D}$ without great decrease in accuracy.

---

**Algorithm A.3** Implementation of a filtering algorithm given an implementation of a generalization algorithm

---

1: **Input**: Training set $S = S_{\text{clean}} \cup S_{\text{adv}}$
   satisfying $|S_{\text{clean}}| = \Omega\left(\varepsilon_{\text{clean}}^{-2}\left(\text{VC}\left(\mathcal{L}_{\mathcal{F}_{\text{adv}}(h^*)}\right) + \log\left(1/\delta\right)\right)\right)$
2: Calculate $\widehat{h} = \text{argmin}_{h \in \mathcal{H}}\mathcal{L}_{\mathcal{F}_{\text{adv}}(h^*)}(h, S)$ and early-return $S$ if $\mathcal{L}_{\mathcal{F}_{\text{adv}}(h^*)}(\widehat{h}, S) \leq C\varepsilon_{\text{clean}}$, for some universal constant $C$
3: Randomly partition $S$ into two equal halves $S_L$ and $S_R$
4: Run the generalizing algorithm to obtain $\widehat{h}_L$ and $\widehat{h}_R$ using training sets $S_L$ and $S_R$, respectively
5: Run $\widehat{h}_L$ on $S_R$ and mark every mistake that $\widehat{h}_L$ makes on $S_R$, and similarly for $\widehat{h}_R$
6: Remove all marked examples to obtain a new training set $S' \subseteq S$
7: **Output**: $S'$ such that $\widehat{h} = \text{argmin}_{h \in \mathcal{H}}\mathcal{L}_{\mathcal{F}_{\text{adv}}(h^*)}(h, S')$ satisfies $\mathcal{L}_{\mathcal{F}_{\text{adv}}(h^*)}(\widehat{h}, \mathcal{D}) \lesssim \varepsilon_{\text{clean}}$ with probability $1 - \delta$

---

We have two cases to consider based on the number of outliers in our training set. Let $m$ be the total number of examples in our training set.

**Case 1** $-\alpha m \leq \max\left\{2/3\varepsilon_{\text{clean}} \cdot \log\left(1/\delta\right), 24\log\left(2/\delta\right)\right\}$   It is easy to see that $\mathcal{L}(h^*, S) \leq \alpha$. Using this, we can write:

$$\mathcal{L}(h^*, S) \leq \alpha$$

$$\frac{2}{3\varepsilon_{\text{clean}} \cdot m} \cdot \log\left(\frac{1}{\delta}\right)$$

$$\lesssim \frac{\varepsilon_{\mathsf{clean}}}{\mathsf{VC}\left(\mathcal{H}\right) + \log\left(1/\delta\right)} \cdot \log\left(\frac{1}{\delta}\right)$$

$$< \varepsilon_{\mathsf{clean}}$$

which implies that we exit the routine via the early-return. From uniform convergence, this implies that with probability $1 - \delta$ over the draws of $S$, we have $\mathcal{L}_{\mathcal{F}_{\mathsf{adv}}(h^*)}\left(\arg\min_{h \in \mathcal{H}} \mathcal{L}_{\mathcal{F}_{\mathsf{adv}}(h^*)}\left(h, S'\right), \mathcal{D}\right) \lesssim \varepsilon_{\mathsf{clean}}$.

In the other case, we write:

$$\mathcal{L}(h^*, S) \leq \alpha$$
$$\leq \frac{24 \log\left(2/\delta\right)}{m}$$
$$\lesssim \frac{\varepsilon_{\mathsf{clean}}^2 \log\left(1/\delta\right)}{\mathsf{VC}\left(\mathcal{H}\right) + \log\left(1/\delta\right)}$$
$$\lesssim \varepsilon_{\mathsf{clean}}^2 \leq \varepsilon_{\mathsf{clean}}$$

and the rest follows from a similar argument.

**Case 2 –** $\alpha m \geq \max\left\{2/3\varepsilon_{\mathsf{clean}} \cdot \log\left(1/\delta\right), 24 \log\left(2/\delta\right)\right\}$  Let $\tau = \delta$; we make this rewrite to help simplify the various failure events.

**Part 1 – Partitioning Doesn't Affect Outlier Balance**  Define indicator random variables $X_i$ such that $X_i$ is 1 if and only if example $i$ ends up in $S_R$. We want to show that:

$$\Pr\left[\sum_{i \in S_{\mathsf{adv}}} X_i \notin [0.5, 1.5]\, \alpha \cdot m/2\right] \leq \tau$$

Although the $X_i$ are not independent, they are negatively associated, so we can still use the Chernoff Bound and the fact that the number of outliers $\alpha m \geq 24 \log\left(2/\tau\right)$:

$$\Pr\left[\sum_{i \in S_{\mathsf{adv}}} X_i \notin [0.5, 1.5]\, \alpha \cdot m/2\right] \leq 2\exp\left(-\frac{\alpha/2 \cdot m \cdot 1/4}{3}\right)$$
$$\leq 2\exp\left(-\frac{\alpha m}{24}\right) \leq \tau$$

Moreover, if $S_L$ has a $[\alpha/2, 3\alpha/2]$ fraction of outliers, then it also follows that $S_R$ has a $[\alpha/2, 3\alpha/2]$ fraction of outliers. Thus, this step succeeds with probability $1 - \tau$.

**Part 2 – Approximately Correctly Marking Points**  We now move onto showing that $\widehat{h}_L$ deletes most outliers from $S_R$ while deleting few clean points. Recall that $\widehat{h}_L$ satisfies $\mathcal{L}_{\mathcal{F}_{\mathsf{adv}}(h^*)}(\widehat{h}_L, \mathcal{D}) \leq \varepsilon_{\mathsf{clean}}$ with probability $1 - \delta$. Thus, we have that $\widehat{h}_L$ labels the outliers as opposite the target label with probability at least $1 - \varepsilon_{\mathsf{clean}}$. Since we have that the number of outliers $\alpha m \geq 2/3\varepsilon_{\mathsf{clean}} \cdot \log\left(1/\tau\right)$, we have from Chernoff Bound (let $X_i$ be the indicator random variable that is 1 when $\widehat{h}_L$ classifies a backdoored example as the target label):

$$\Pr\left[\sum_{i \in S_{\mathsf{adv}} \cap S_R} X_i \geq 2 \cdot \left(\varepsilon_{\mathsf{clean}} \cdot \frac{3}{2}\alpha m\right)\right] \leq \exp\left(-\varepsilon_{\mathsf{clean}} \cdot \frac{3}{2}\alpha m\right) \leq \tau$$

Thus, with probability $1 - 2\tau$, we mark all but at most $\varepsilon_{\mathsf{clean}} \cdot 6\alpha m$ outliers across both $S_R$ and $S_L$; since we impose that $\alpha \lesssim 1$, we have that we delete all but a $c\varepsilon_{\mathsf{clean}}$ fraction of outliers for some universal constant $c$.

It remains to show that we don't delete too many good points. Since $\widehat{h}_L$ has true error at most $\varepsilon_{\mathsf{clean}}$ and using the fact that $m(1 - \alpha/2) \geq m(1 - \alpha) \geq m\alpha \geq \frac{2 \log(1/\tau)}{\varepsilon_{\mathsf{clean}}}$, from the Chernoff Bound, we have (let $X_i$ be the indicator random variable that is 1 when $\widehat{h}_L$ misclassifies a clean example):

$$\Pr\left[\sum_{i \in S_{\mathsf{clean}} \cap S_R} X_i \geq 2 \cdot \left(\varepsilon_{\mathsf{clean}} \cdot (1 - \alpha/2) \cdot \frac{m}{2}\right)\right] \leq \exp\left(-\varepsilon_{\mathsf{clean}} \cdot (1 - \alpha/2) \cdot \frac{m}{2}\right) \leq \tau$$

From a union bound over the runs of $\widehat{h}_L$ and $\widehat{h}_R$, we have that with probability $1 - 2\tau$, we mark at most $2m\varepsilon_{\mathsf{clean}} \cdot (1 - \alpha/2) \leq 2m\varepsilon_{\mathsf{clean}}$ clean points for deletion. From a union bound, we have that this whole step succeeds with probability $1 - 4\tau - 2\delta$.

**Part 3 – There Exists a Low-Error Classifier**  At this stage, we have a training set $S'$ that has at least $m(1 - 2\varepsilon_{\text{clean}})$ clean points and at most $\varepsilon_{\text{clean}} \cdot 6\alpha m$ outliers. Recall that $h^*$ incurs robust loss on none of the clean points and incurs robust loss on every outlier. This implies that $h^*$ has empirical robust loss at most:

$$\frac{\varepsilon_{\text{clean}} \cdot 6\alpha m}{m(1 - 2\varepsilon_{\text{clean}})} = \frac{6\alpha\varepsilon_{\text{clean}}}{1 - 2\varepsilon_{\text{clean}}} \leq 2\varepsilon_{\text{clean}}$$

where we use the fact that we pick $\varepsilon_{\text{clean}} \leq 1/10 < 1/4$ and $\alpha \leq 1/6$. From this, it follows that $\widehat{h} = \text{argmin}_{h \in \mathcal{H}} \mathcal{L}_{\mathcal{F}_{\text{adv}}(h^*)}(h, S')$ satisfies $\mathcal{L}_{\mathcal{F}_{\text{adv}}(h^*)}(\widehat{h}, S') \leq 2\varepsilon_{\text{clean}}$.

**Part 4 – Generalizing from $S'$ to $\mathcal{D}$**  We now have to argue that $\mathcal{L}_{\mathcal{F}_{\text{adv}}(h^*)}(\widehat{h}, S') \leq 2\varepsilon_{\text{clean}}$ implies $\mathcal{L}_{\mathcal{F}_{\text{adv}}(h^*)}(\widehat{h}, \mathcal{D}) \lesssim \varepsilon_{\text{clean}}$. Recall that we have drawn enough samples to achieve uniform convergence (see [14] and [26]); in particular, assuming that our previous steps succeeded in removing very few points from $S_{\text{clean}}$, then for all $h \in \mathcal{H}$, we have with probability $1 - \delta$:

$$\left| \mathcal{L}_{\mathcal{F}_{\text{adv}}(h^*)}(h, \mathcal{D}) - \mathcal{L}_{\mathcal{F}_{\text{adv}}(h^*)}(h, S_{\text{clean}}) \right| \leq \varepsilon_{\text{clean}}$$

Observe that we have deleted at most $m \cdot 2\varepsilon_{\text{clean}}$ points from $S_{\text{clean}}$. Let $S'_{\text{clean}} := S' \cap S_{\text{clean}}$ (i.e., the surviving members of $S_{\text{clean}}$ from our filtering procedure). We start with the following claim.

**Claim 34.** *The following holds for all $h \in \mathcal{H}$:*

$$\left| \mathcal{L}_{\mathcal{F}_{\text{adv}}(h^*)}(h, S_{\text{clean}}) - \mathcal{L}_{\mathcal{F}_{\text{adv}}(h^*)}(h, S'_{\text{clean}}) \right| < 3\varepsilon_{\text{clean}}$$

*Proof.* Recall that in the proof of Theorem 17, we showed that for positive numbers $a, b, c$ we have:

$$\max\left\{0, \frac{a - c}{b - c}\right\}, \min\left\{1, \frac{a}{b - c}\right\} \in \left[\frac{a}{b} \pm \frac{c}{b}\right]$$

Now, let $a$ denote the number of samples from $S_{\text{clean}}$ that $h$ incurs robust loss on, let $b$ be the total number of samples from $S_{\text{clean}}$, and let $c$ be the number of samples our filtering procedure deletes from $S_{\text{clean}}$. It is easy to see that $a/b$ corresponds $\mathcal{L}_{\mathcal{F}_{\text{adv}}(h^*)}(h, S_{\text{clean}})$ and that $\mathcal{L}_{\mathcal{F}_{\text{adv}}(h^*)}(h, S'_{\text{clean}}) \in [\max\{0, (a-c)/(b-c)\}, \min\{1, a/(b-c)\}]$. From our argument above, this means that we must have:

$$\mathcal{L}_{\mathcal{F}_{\text{adv}}(h^*)}(h, S'_{\text{clean}}) \in \left[\mathcal{L}_{\mathcal{F}_{\text{adv}}(h^*)}(h, S_{\text{clean}}) \pm \frac{2\varepsilon_{\text{clean}} m}{(1 - \alpha)m}\right]$$

Finally:

$$\frac{2\varepsilon_{\text{clean}} m}{(1 - \alpha)m} = \frac{2\varepsilon_{\text{clean}}}{(1 - \alpha)} \leq \frac{2\varepsilon_{\text{clean}}}{5/6} < 3\varepsilon_{\text{clean}}$$

and we're done. $\qquad\square$

We now use our claim and triangle inequality to write:

$$\left| \mathcal{L}_{\mathcal{F}_{\text{adv}}(h^*)}(h, S'_{\text{clean}}) - \mathcal{L}_{\mathcal{F}_{\text{adv}}(h^*)}(h, \mathcal{D}) \right| \leq \left| \mathcal{L}_{\mathcal{F}_{\text{adv}}(h^*)}(h, S_{\text{clean}}) - \mathcal{L}_{\mathcal{F}_{\text{adv}}(h^*)}(h, S'_{\text{clean}}) \right| +$$
$$\left| \mathcal{L}_{\mathcal{F}_{\text{adv}}(h^*)}(h, \mathcal{D}) - \mathcal{L}_{\mathcal{F}_{\text{adv}}(h^*)}(h, S_{\text{clean}}) \right|$$
$$< 4\varepsilon_{\text{clean}}$$

Next, consider some $\widehat{h}$ satisfying $\mathcal{L}_{\mathcal{F}_{\text{adv}}(h^*)}(\widehat{h}, S') \leq 2\varepsilon_{\text{clean}}$ (which must exist, as per our argument in Part 3), and observe that:

$$\mathcal{L}_{\mathcal{F}_{\text{adv}}(h^*)}(\widehat{h}, S') \geq (1 - 2\varepsilon_{\text{clean}})\mathcal{L}_{\mathcal{F}_{\text{adv}}(h^*)}(\widehat{h}, S' \cap S_{\text{clean}}) + 2\varepsilon_{\text{clean}}\mathcal{L}_{\mathcal{F}_{\text{adv}}(h^*)}(\widehat{h}, S' \cap S_{\text{adv}})$$
$$\geq (1 - 2\varepsilon_{\text{clean}})\mathcal{L}_{\mathcal{F}_{\text{adv}}(h^*)}(\widehat{h}, S'_{\text{clean}})$$
$$\Rightarrow \mathcal{L}_{\mathcal{F}_{\text{adv}}(h^*)}(\widehat{h}, S'_{\text{clean}}) \leq \frac{2\varepsilon_{\text{clean}}}{1 - 2\varepsilon_{\text{clean}}} = 2\varepsilon_{\text{clean}}\left(\frac{1}{1 - 2\varepsilon_{\text{clean}}}\right) \leq \frac{5\varepsilon_{\text{clean}}}{2}$$

We now use the fact that $\left| \mathcal{L}_{\mathcal{F}_{\text{adv}}(h^*)}(h, S'_{\text{clean}}) - \mathcal{L}_{\mathcal{F}_{\text{adv}}(h^*)}(h, \mathcal{D}) \right| < 4\varepsilon_{\text{clean}}$ to arrive at the conclusion that $\mathcal{L}_{\mathcal{F}_{\text{adv}}(h^*)}(h, \mathcal{D}) < 13/2 \cdot \varepsilon_{\text{clean}}$, which is what we wanted to show.

The constants in the statement of Theorem 18 follow from setting $\tau = \delta$. $\qquad\square$

# B   Numerical Trials

In this section, we present a practical use case for Theorem 14 (Appendix Theorem 30).

Recall that, at a high level, Theorem 14 states that under certain assumptions, minimizing robust loss on the corrupted training set will either:

1.  Result in a low robust loss, which will imply from uniform convergence that the resulting classifier is robust to adversarial (and therefore backdoor) perturbations.

2.  Result in a high robust loss, which will be noticeable at training time.

This suggests that practitioners can use adversarial training on a training set which may be backdoored and use the resulting robust loss value to make a decision about whether to deploy the classifier. To empirically validate this approach, we run this procedure (i.e., some variant of Algorithm A.1) on the MNIST handwritten digit classification task[6](see [2]). Here, the learner wishes to recover a neural network robust to small $\ell_\infty$ perturbations and where the adversary is allowed to make a small $\ell_\infty$-norm watermark.

**Disclaimers**   As far as we are aware, the MNIST dataset does not contain personally identifiable information or objectionable content. The MNIST dataset is made available under the terms of the Creative Commons Attribution-Share Alike 3.0 License.

**Reproducibility**   We have included all the code to generate these results in the supplementary material.   Our code can be found at `https://github.com/narenmanoj/mnist-adv-training`.[7]. Our code is tested and working with TensorFlow 2.4.1, CUDA 11.0, NVIDIA RTX 2080Ti, and the Google Colab GPU runtime.

## B.1   MNIST Using Neural Networks

### B.1.1   Scenario

Recall that the MNIST dataset consists of 10 classes, where each corresponds to a handwritten digit in $\{0, \dots, 9\}$. The classification task here is to recover a classifier that, upon receiving an image of a handwritten digit, correctly identifies which digit is present in the image.

In our example use case, an adversary picks a target label $t \in \{0, \dots, 9\}$ and a small additive watermark. If the true classifier is $h^*(x)$, then the adversary wants the learner to find a classifier $\widehat{h}$ maximizing $\Pr_{x \sim \mathcal{D}|h^*(x) \neq t} \left[ \widehat{h}(x) = t \right]$. In other words, this can be seen as a "many-to-one" attack, where the adversary is corrupting examples whose labels are not $t$ in order to induce a classification of $t$. The adversary is allowed to inject some number of examples into the training set such that the resulting fraction of corrupted examples in the training set is at most $\alpha$.

We will experimentally demonstrate that the learner can use the intuition behind Theorem 14 (Appendix Theorem 30) to either recover a reasonably robust classifier or detect the presence of significant corruptions in the training set. Specifically, the learner can optimize a proxy for the robust loss via adversarial training using $\ell_\infty$ bounded adversarial examples, as done by [11].

**Instantiation of Relevant Problem Parameters**   Let $\mathcal{H}$ be the set of neural networks with architecture as shown in Table 1. Let $\mathcal{X}$ be the set of images of handwritten digits; we represent these as vectors in $[0, 1]^{784}$. Define $\mathcal{F}_{\mathsf{adv}}$ below:

$$\{\mathsf{patch}\,(x)  \; : \;  \|x - \mathsf{patch}\,(x)\|_\infty \leq 0.3 \text{ and } \mathsf{patch}\,(x) - x = \mathsf{pattern}\}$$

where pattern is the shape of the backdoor (we use an "X" shape in the top left corner of the image, inspired by [17]). We let the maximum $\ell_\infty$ perturbation be at most 0.3 since this parameter has been historically used in training and evaluating robust networks on MNIST (see [11]). In our setup, we

---

[6]We select MNIST because one can achieve a reasonably robust classifier on the clean version of the dataset. This helps us underscore the difference between the robust loss at train time with and without backdoors in the training set. Moreover, this allows us to explore a setting where our assumptions in Theorem 14 might not hold – in particular, it's not clear that we have enough data to attain uniform convergence for the binary loss and $\mathcal{L}_{\mathcal{F}_{\mathsf{adv}}(h^*)}$, and it's not clear how to efficiently minimize $\mathcal{L}_{\mathcal{F}_{\mathsf{adv}}(h^*)}$.

[7]Some of our code is derived from the GitHub repositories `https://github.com/MadryLab/backdoor_data_poisoning` and `https://github.com/skmda37/Adversarial_Machine_Learning_Tensorflow`.

demonstrate that these parameters suffice to yield a successful backdoor attack on a vanilla training procedure (described in greater detail in a subsequent paragraph).

Although it is not clear how to efficiently exactly calculate and minimize $\mathcal{L}_{\mathcal{F}_{\text{adv}}(h^*)}$, we will approximate $\mathcal{L}_{\mathcal{F}_{\text{adv}}(h^*)}$ by calculating $\ell_\infty$-perturbed adversarial examples using a Projected Gradient Descent (PGD) attack. To minimize $\mathcal{L}_{\mathcal{F}_{\text{adv}}(h^*)}$, we use adversarial training as described in [11]. Generating Table 3 takes roughly 155 minutes using our implementation of this procedure with TensorFlow 2.4.1 running on the GPU runtime freely available via Google Colab. We list all our relevant optimization and other experimental parameters in Table 2.

Table 1: Neural network architecture used in experiments. We implemented this architecture using the Keras API of TensorFlow 2.4.1.

| Layer | Parameters |
|---|---|
| `Conv2D` | `filters=32,kernel_size=(3,3),activation='relu'` |
| `MaxPooling2D` | `pool_size=(2,2)` |
| `Conv2D` | `filters=64,kernel_size=(3,3),activation='relu'` |
| `Flatten` | |
| `Dense` | `units=1024,activation='relu'` |
| `Dense` | `units=10,activation='softmax'` |

Table 2: Experimental hyperparameters. We made no effort to optimize these hyperparameters; indeed, many of these are simply the default arguments for the respective TensorFlow functions.

| Property | Details |
|---|---|
| Epochs | 2 |
| Validation Split | None |
| Batch Size | 32 |
| Loss | Sparse Categorical Cross Entropy |
| Optimizer | RMSProp (step size = 0.001, $\rho = 0.9$, momentum = 0, $\varepsilon = 10^{-7}$) |
| NumPy Random Seed | 4321 |
| TensorFlow Random Seed | 1234 |
| PGD Attack | $\varepsilon = 0.3$, step size = 0.01, iterations = 40, restarts = 10 |

**Optimization Details**   See Table 2 for all relevant hyperparameters and see Table 1 for the architecture we use.

For the "Vanilla Training" procedure, we use no adversarial training and simply use our optimizer to minimize our loss directly. For the "PGD-Adversarial Training" procedure, we use adversarial training with a PGD adversary.

In our implementation of adversarial training, we compute adversarial examples for each image in each batch using the PGD attack and we minimize our surrogate loss on this new batch. This is sufficient to attain a classifier with estimated robust loss of around 0.08 on an uncorrupted training set.

### B.1.2   Goals and Evaluation Methods

We want to observe the impact of adding backdoor examples and the impact of running adversarial training on varied values of $\alpha$ (the fraction of the training set that is corrupted).

To do so, we fix a value for $\alpha$ and a target label $t$ and inject enough backdoor examples such that exactly an $\alpha$ fraction of the resulting training set contains corrupted examples. Then, we evaluate the train and test robust losses on the training set with and without adversarial training to highlight the difference in robust loss observable to the learner. As sanity checks, we also include binary losses and test set metrics. For the full set of metrics we collect, see Table 3.

To avoid out-of-memory issues when computing the robust loss on the full training set (roughly 60000 training examples and their adversarial examples), we sample 5000 training set examples uniformly at random from the full training set and compute the robust loss on these examples. By Hoeffding's Inequality (see [18]), this means that with probability 0.99 over the choice of the subsampled training set, the difference between our reported statistic and its population value is at most $\sim 0.02$.

### B.1.3 Results and Discussion

Table 3: Results with MNIST with a target label $t = 0$ and backdoor pattern "X." In each cell, the top number represents the respective value when the network was trained without any kind of robust training, and the bottom number represents the respective value when the network was trained using adversarial training as per [11]. For example, at $\alpha = 0.05$, for Vanilla Training, the training $0 - 1$ loss is only 0.01, but the training robust loss is 1.00, whereas for PGD-Adversarial Training, the training $0 - 1$ loss is 0.07 and the training robust loss is 0.13. The Backdoor Success Rate is our estimate of $\Pr_{x \sim \mathcal{D} || y \neq t} [\text{patch}(x) = t]$, which may be less than the value of the robust loss.

| $\alpha$ | | 0.00 | 0.05 | 0.15 | 0.20 | 0.30 |
|---|---|---|---|---|---|---|
| Training $0 - 1$ Loss | Vanilla Training | 0.01 | 0.01 | 0.01 | 0.01 | 0.01 |
| | PGD-Adversarial Training | 0.02 | 0.07 | 0.17 | 0.22 | 0.33 |
| Training Robust Loss | Vanilla Training | 1.00 | 1.00 | 1.00 | 1.00 | 1.00 |
| | PGD-Adversarial Training | 0.09 | 0.13 | 0.24 | 0.27 | 0.41 |
| Testing $0 - 1$ Loss | Vanilla Training | 0.01 | 0.01 | 0.01 | 0.02 | 0.01 |
| | PGD-Adversarial Training | 0.02 | 0.03 | 0.03 | 0.03 | 0.06 |
| Testing Robust Loss | Vanilla Training | 1.00 | 1.00 | 1.00 | 1.00 | 1.00 |
| | PGD-Adversarial Training | 0.09 | 0.09 | 0.11 | 0.10 | 0.19 |
| Backdoor Success Rate | Vanilla Training | 0.00 | 1.00 | 1.00 | 1.00 | 1.00 |
| | PGD-Adversarial Training | 0.00 | 0.00 | 0.01 | 0.00 | 0.05 |

See Table 3 for sample results from our trials. Over runs of the same experiment with varied target labels $t$, we attain similar results; see Section B.1.4 for the full results. We now discuss the key takeaways from this numerical trial.

**Training Robust Loss Increases With $\alpha$**  Observe that our proxy for $\mathcal{L}_{\mathcal{F}_{\text{adv}}(h^*)}(\widehat{h}, S)$ increases as $\alpha$ increases. This is consistent with the intuition from Theorem 14 in that a highly corrupted training set is unlikely to have low robust loss. Hence, if the learner expects a reasonably low robust loss and fails to observe this during training, then the learner can reject the training set, particularly at high $\alpha$.

**Smaller $\alpha$ and Adversarial Training Defeats Backdoor**  On the other hand, notice that at smaller values of $\alpha$ (particularly $\alpha \leq 0.20$), the learner can still recover a classifier with minimal decrease in robust accuracy. Furthermore, there is not an appreciable decrease in natural accuracy either when using adversarial training on a minimally corrupted training set. Interestingly, even at large $\alpha$, the test-time robust loss and binary losses are not too high when adversarial training was used. Furthermore, the test-time robust loss attained at $\alpha > 0$ is certainly better than that obtained when adversarial training is not used, even at $\alpha = 0$. Hence, although the practitioner cannot certify that the learned model is robust without a clean validation set, the learned model does tend to be fairly robust.

**Backdoor Is Successful With Vanilla Training**  Finally, as a sanity check, notice that when we use vanilla training, the backdoor trigger induces a targeted misclassification very reliably, even at $\alpha = 0.05$. Furthermore, the training and testing error on clean data is very low, which indicates that the learner would have failed to detect the fact that the model had been corrupted had they checked only the training and testing errors before deployment.

**Prior Empirical Work**  The work of [31] empirically shows the power of data augmentation in defending against backdoored training sets. Although their implementation of data augmentation is different from ours[8], their work still demonstrates that attempting to minimize some proxy for the robust loss can lead to a classifier robust to backdoors at test time. However, our evaluation also demonstrates that classifiers trained using adversarial training can be robust against test-time adversarial attacks, in addition to being robust to train-time backdoor attacks. Furthermore, our empirical results indicate that the train-time robust loss can serve as a good indicator for whether a significant number of backdoors are in the training set.

---

[8]Observe that our implementation of adversarial training can be seen as a form of adaptive data augmentation.

### B.1.4 Results For All Target Labels

Here, we present tables of the form of Table 3 for all choices of target label $t \in \{0, \ldots, 9\}$. Notice that the key takeaways remain the same across all target labels.

Table 4: Results with MNIST with a target label $t = 0$ and backdoor pattern "X."

| $\alpha$ | | 0.00 | 0.05 | 0.15 | 0.20 | 0.30 |
|---|---|---|---|---|---|---|
| Training $0 - 1$ Loss | Vanilla Training | 0.01 | 0.01 | 0.01 | 0.01 | 0.01 |
| | PGD-Adversarial Training | 0.02 | 0.07 | 0.17 | 0.22 | 0.33 |
| Training Robust Loss | Vanilla Training | 1.00 | 1.00 | 1.00 | 1.00 | 1.00 |
| | PGD-Adversarial Training | 0.09 | 0.13 | 0.24 | 0.27 | 0.41 |
| Testing $0 - 1$ Loss | Vanilla Training | 0.01 | 0.01 | 0.01 | 0.02 | 0.01 |
| | PGD-Adversarial Training | 0.02 | 0.03 | 0.03 | 0.03 | 0.06 |
| Testing Robust Loss | Vanilla Training | 1.00 | 1.00 | 1.00 | 1.00 | 1.00 |
| | PGD-Adversarial Training | 0.09 | 0.09 | 0.11 | 0.10 | 0.19 |
| Backdoor Success Rate | Vanilla Training | 0.00 | 1.00 | 1.00 | 1.00 | 1.00 |
| | PGD-Adversarial Training | 0.00 | 0.00 | 0.01 | 0.00 | 0.05 |

Table 5: Results with MNIST with a target label $t = 1$ and backdoor pattern "X."

| $\alpha$ | | 0.00 | 0.05 | 0.15 | 0.20 | 0.30 |
|---|---|---|---|---|---|---|
| Training $0 - 1$ Loss | Vanilla Training | 0.01 | 0.01 | 0.01 | 0.01 | 0.01 |
| | PGD-Adversarial Training | 0.02 | 0.07 | 0.17 | 0.23 | 0.32 |
| Training Robust Loss | Vanilla Training | 1.00 | 1.00 | 1.00 | 1.00 | 1.00 |
| | PGD-Adversarial Training | 0.08 | 0.12 | 0.23 | 0.32 | 0.38 |
| Testing $0 - 1$ Loss | Vanilla Training | 0.01 | 0.01 | 0.01 | 0.01 | 0.01 |
| | PGD-Adversarial Training | 0.02 | 0.02 | 0.03 | 0.04 | 0.05 |
| Testing Robust Loss | Vanilla Training | 1.00 | 1.00 | 1.00 | 1.00 | 1.00 |
| | PGD-Adversarial Training | 0.09 | 0.08 | 0.11 | 0.13 | 0.14 |
| Backdoor Success Rate | Vanilla Training | 0.00 | 1.00 | 1.00 | 1.00 | 1.00 |
| | PGD-Adversarial Training | 0.00 | 0.00 | 0.00 | 0.02 | 0.03 |

Table 6: Results with MNIST with a target label $t = 2$ and backdoor pattern "X."

| $\alpha$ | | 0.00 | 0.05 | 0.15 | 0.20 | 0.30 |
|---|---|---|---|---|---|---|
| Training $0 - 1$ Loss | Vanilla Training | 0.01 | 0.01 | 0.01 | 0.01 | 0.00 |
| | PGD-Adversarial Training | 0.02 | 0.07 | 0.17 | 0.22 | 0.32 |
| Training Robust Loss | Vanilla Training | 1.00 | 1.00 | 1.00 | 1.00 | 1.00 |
| | PGD-Adversarial Training | 0.08 | 0.13 | 0.23 | 0.28 | 0.38 |
| Testing $0 - 1$ Loss | Vanilla Training | 0.01 | 0.02 | 0.01 | 0.02 | 0.01 |
| | PGD-Adversarial Training | 0.02 | 0.03 | 0.03 | 0.03 | 0.05 |
| Testing Robust Loss | Vanilla Training | 1.00 | 1.00 | 1.00 | 1.00 | 1.00 |
| | PGD-Adversarial Training | 0.09 | 0.09 | 0.10 | 0.10 | 0.14 |
| Backdoor Success Rate | Vanilla Training | 0.00 | 1.00 | 1.00 | 1.00 | 1.00 |
| | PGD-Adversarial Training | 0.00 | 0.00 | 0.00 | 0.01 | 0.04 |

Table 7: Results with MNIST with a target label $t = 3$ and backdoor pattern "X."

| $\alpha$ | | 0.00 | 0.05 | 0.15 | 0.20 | 0.30 |
|---|---|---|---|---|---|---|
| Training $0 - 1$ Loss | Vanilla Training | 0.01 | 0.01 | 0.01 | 0.01 | 0.01 |
| | PGD-Adversarial Training | 0.02 | 0.07 | 0.18 | 0.23 | 0.32 |
| Training Robust Loss | Vanilla Training | 1.00 | 1.00 | 1.00 | 1.00 | 1.00 |
| | PGD-Adversarial Training | 0.08 | 0.13 | 0.23 | 0.28 | 0.38 |
| Testing $0 - 1$ Loss | Vanilla Training | 0.01 | 0.01 | 0.01 | 0.02 | 0.02 |
| | PGD-Adversarial Training | 0.02 | 0.02 | 0.03 | 0.04 | 0.05 |
| Testing Robust Loss | Vanilla Training | 1.00 | 1.00 | 1.00 | 1.00 | 1.00 |
| | PGD-Adversarial Training | 0.09 | 0.09 | 0.11 | 0.11 | 0.13 |
| Backdoor Success Rate | Vanilla Training | 0.00 | 1.00 | 1.00 | 1.00 | 1.00 |
| | PGD-Adversarial Training | 0.00 | 0.01 | 0.00 | 0.01 | 0.03 |

Table 8: Results with MNIST with a target label $t = 4$ and backdoor pattern "X."

| $\alpha$ | | 0.00 | 0.05 | 0.15 | 0.20 | 0.30 |
|---|---|---|---|---|---|---|
| Training $0 - 1$ Loss | Vanilla Training | 0.01 | 0.01 | 0.01 | 0.01 | 0.01 |
| | PGD-Adversarial Training | 0.02 | 0.07 | 0.17 | 0.22 | 0.32 |
| Training Robust Loss | Vanilla Training | 1.00 | 1.00 | 1.00 | 1.00 | 1.00 |
| | PGD-Adversarial Training | 0.08 | 0.13 | 0.24 | 0.27 | 0.42 |
| Testing $0 - 1$ Loss | Vanilla Training | 0.01 | 0.01 | 0.01 | 0.01 | 0.01 |
| | PGD-Adversarial Training | 0.02 | 0.02 | 0.03 | 0.03 | 0.05 |
| Testing Robust Loss | Vanilla Training | 1.00 | 1.00 | 1.00 | 1.00 | 1.00 |
| | PGD-Adversarial Training | 0.08 | 0.09 | 0.11 | 0.10 | 0.15 |
| Backdoor Success Rate | Vanilla Training | 0.00 | 1.00 | 1.00 | 1.00 | 1.00 |
| | PGD-Adversarial Training | 0.00 | 0.00 | 0.01 | 0.01 | 0.04 |

Table 9: Results with MNIST with a target label $t = 5$ and backdoor pattern "X."

| $\alpha$ | | 0.00 | 0.05 | 0.15 | 0.20 | 0.30 |
|---|---|---|---|---|---|---|
| Training $0 - 1$ Loss | Vanilla Training | 0.01 | 0.01 | 0.01 | 0.01 | 0.01 |
| | PGD-Adversarial Training | 0.02 | 0.07 | 0.17 | 0.22 | 0.33 |
| Training Robust Loss | Vanilla Training | 1.00 | 1.00 | 1.00 | 1.00 | 1.00 |
| | PGD-Adversarial Training | 0.07 | 0.13 | 0.23 | 0.28 | 0.41 |
| Testing $0 - 1$ Loss | Vanilla Training | 0.01 | 0.01 | 0.01 | 0.02 | 0.02 |
| | PGD-Adversarial Training | 0.02 | 0.03 | 0.03 | 0.03 | 0.06 |
| Testing Robust Loss | Vanilla Training | 1.00 | 1.00 | 1.00 | 1.00 | 1.00 |
| | PGD-Adversarial Training | 0.08 | 0.09 | 0.11 | 0.10 | 0.16 |
| Backdoor Success Rate | Vanilla Training | 0.00 | 1.00 | 1.00 | 1.00 | 1.00 |
| | PGD-Adversarial Training | 0.00 | 0.00 | 0.01 | 0.01 | 0.05 |

Table 10: Results with MNIST with a target label $t = 6$ and backdoor pattern "X."

| $\alpha$ | | 0.00 | 0.05 | 0.15 | 0.20 | 0.30 |
|---|---|---|---|---|---|---|
| Training $0 - 1$ Loss | Vanilla Training | 0.01 | 0.01 | 0.01 | 0.01 | 0.01 |
| | PGD-Adversarial Training | 0.02 | 0.07 | 0.17 | 0.22 | 0.33 |
| Training Robust Loss | Vanilla Training | 1.00 | 1.00 | 1.00 | 1.00 | 1.00 |
| | PGD-Adversarial Training | 0.08 | 0.12 | 0.24 | 0.27 | 0.40 |
| Testing $0 - 1$ Loss | Vanilla Training | 0.01 | 0.02 | 0.01 | 0.01 | 0.01 |
| | PGD-Adversarial Training | 0.02 | 0.03 | 0.03 | 0.03 | 0.06 |
| Testing Robust Loss | Vanilla Training | 1.00 | 1.00 | 1.00 | 1.00 | 1.00 |
| | PGD-Adversarial Training | 0.09 | 0.09 | 0.12 | 0.10 | 0.16 |
| Backdoor Success Rate | Vanilla Training | 0.00 | 1.00 | 1.00 | 1.00 | 1.00 |
| | PGD-Adversarial Training | 0.00 | 0.00 | 0.01 | 0.01 | 0.04 |

Table 11: Results with MNIST with a target label $t = 7$ and backdoor pattern "X."

| $\alpha$ | | 0.00 | 0.05 | 0.15 | 0.20 | 0.30 |
|---|---|---|---|---|---|---|
| Training $0 - 1$ Loss | Vanilla Training | 0.01 | 0.01 | 0.01 | 0.01 | 0.01 |
| | PGD-Adversarial Training | 0.02 | 0.07 | 0.18 | 0.22 | 0.32 |
| Training Robust Loss | Vanilla Training | 1.00 | 1.00 | 1.00 | 1.00 | 1.00 |
| | PGD-Adversarial Training | 0.07 | 0.12 | 0.25 | 0.29 | 0.39 |
| Testing $0 - 1$ Loss | Vanilla Training | 0.01 | 0.01 | 0.01 | 0.02 | 0.01 |
| | PGD-Adversarial Training | 0.02 | 0.03 | 0.03 | 0.03 | 0.04 |
| Testing Robust Loss | Vanilla Training | 1.00 | 1.00 | 1.00 | 1.00 | 1.00 |
| | PGD-Adversarial Training | 0.08 | 0.08 | 0.11 | 0.10 | 0.13 |
| Backdoor Success Rate | Vanilla Training | 0.00 | 1.00 | 1.00 | 1.00 | 1.00 |
| | PGD-Adversarial Training | 0.00 | 0.00 | 0.00 | 0.00 | 0.03 |

Table 12: Results with MNIST with a target label $t = 8$ and backdoor pattern "X."

| $\alpha$ | | 0.00 | 0.05 | 0.15 | 0.20 | 0.30 |
|---|---|---|---|---|---|---|
| Training $0 - 1$ Loss | Vanilla Training | 0.01 | 0.01 | 0.01 | 0.01 | 0.01 |
| | PGD-Adversarial Training | 0.02 | 0.07 | 0.17 | 0.22 | 0.32 |
| Training Robust Loss | Vanilla Training | 1.00 | 1.00 | 1.00 | 1.00 | 1.00 |
| | PGD-Adversarial Training | 0.08 | 0.14 | 0.23 | 0.28 | 0.41 |
| Testing $0 - 1$ Loss | Vanilla Training | 0.01 | 0.01 | 0.01 | 0.01 | 0.01 |
| | PGD-Adversarial Training | 0.02 | 0.03 | 0.03 | 0.03 | 0.05 |
| Testing Robust Loss | Vanilla Training | 1.00 | 1.00 | 1.00 | 1.00 | 1.00 |
| | PGD-Adversarial Training | 0.08 | 0.09 | 0.11 | 0.10 | 0.17 |
| Backdoor Success Rate | Vanilla Training | 0.00 | 1.00 | 1.00 | 1.00 | 1.00 |
| | PGD-Adversarial Training | 0.00 | 0.00 | 0.01 | 0.01 | 0.05 |

Table 13: Results with MNIST with a target label $t = 9$ and backdoor pattern "X."

| $\alpha$ | | 0.00 | 0.05 | 0.15 | 0.20 | 0.30 |
|---|---|---|---|---|---|---|
| Training $0 - 1$ Loss | Vanilla Training | 0.01 | 0.01 | 0.01 | 0.01 | 0.01 |
| | PGD-Adversarial Training | 0.02 | 0.07 | 0.17 | 0.22 | 0.33 |
| Training Robust Loss | Vanilla Training | 1.00 | 1.00 | 1.00 | 1.00 | 1.00 |
| | PGD-Adversarial Training | 0.08 | 0.13 | 0.23 | 0.29 | 0.43 |
| Testing $0 - 1$ Loss | Vanilla Training | 0.01 | 0.01 | 0.01 | 0.01 | 0.01 |
| | PGD-Adversarial Training | 0.02 | 0.03 | 0.03 | 0.04 | 0.06 |
| Testing Robust Loss | Vanilla Training | 1.00 | 1.00 | 1.00 | 1.00 | 1.00 |
| | PGD-Adversarial Training | 0.09 | 0.10 | 0.11 | 0.11 | 0.20 |
| Backdoor Success Rate | Vanilla Training | 0.00 | 1.00 | 1.00 | 1.00 | 1.00 |
| | PGD-Adversarial Training | 0.01 | 0.01 | 0.01 | 0.01 | 0.06 |