# OpenReview forum: "Excess Capacity and Backdoor Poisoning"
_NeurIPS.cc/2021/Conference — NeurIPS 2021 Spotlight_

### Official Review · Reviewer_7uMB · 2021-07-08

**Rating:** 7
**Confidence:** 4

**Summary:**

The paper offers a theoretical framework to analyze and understand backdoor attacks. They define a notion of memorization capacity to characterize vulnerability of the learning problem to backdoor attacks. Under this framework they give examples of learning problems where explicit backdoors can be constructed easily and problems which are intrinsically robust to backdoors. Further, they also show that under certain assumptions, backdoor filtering and robust generalization are nearly equivalent. Which suggests, it is sufficient to design backdoor filtering algorithms.

**Limitations And Societal Impact:**

One key limitation is the current work is focused heavily on overparameterized linear classifiers, which are a good starting point. Would have been nice to generalize these results to non-linear/ deep net classifiers.

**Main Review:**

Recently, lots of works have shown vulnerability of deep neural networks to backdoor attacks. However most of these works have been empirical in nature and based on designing/ detecting the attack and proper theoretical understanding of backdoor attacks is lacking. This paper tries to fill this gap by offering a nice theoretical framework to understand whether the given learning setting is vulnerable to backdoor attacks or not and how we can understand it quantitatively. Their notion of memorization capacity and results are definitely valuable in this context.  Although they have nice set of results, I find this paper lacking in clarity. Intuitions and proof sketches are expected at least for the main theorems in the main paper. It would have been better, if the paper elaborated more on the connection between excess capacity and vulnerability to backdoors.

**Time Spent Reviewing:**

3

---

> ### Author Response · Authors · 2021-08-10
> **Response to Official Review of Paper4597 by Reviewer 7uMB**
>
> Thank you for the positive review and the suggestions. Please see our responses to your comments below.
>
> _"Intuitions and proof sketches are expected at least for the main theorems in the main paper."_
>
> Thank you for pointing this out. We originally omitted proof sketches in the main paper in the interest of brevity, but we will add intuition for the proofs of the results stated in the abstract in the main paper for the camera-ready version.
>
> _"It would have been better, if the paper elaborated more on the connection between excess capacity and vulnerability to backdoors."_
>
> Theorems 9 and 10 make explicit the connection between memorization capacity and backdoor vulnerability. However, we now realize that confusion may arise from our interchangeable use of the terms "excess capacity" and "memorization capacity"; we make no distinction between these phrases. We will clarify that we take these phrases to mean the same thing in the camera-ready version of the paper. Furthermore, we will re-emphasize the significance of Theorems 9 and 10 throughout the paper. Specifically, we will pay special attention to this connection in the experimental sub-subsection (Appendix B).
>
> _"Would have been nice to generalize these results to non-linear/ deep net classifiers."_
>
> Theorems 9 and 10 address the backdoor data poisoning problem in greater generality. In particular, we only require the assumptions in lines 99-104 and that the VC dimension of the hypothesis class in question be finite for these results to hold. However, it is true that some of our more specific results (e.g. the fact that a random additive watermark serves as a suitable backdoor) are discussed only in the context of overparameterized linear classifiers. As a future direction, it will be very interesting to find other learning problems for which such "limited information" attacks can also be successful. As you point out, there may be hope to show something similar for neural network classifiers.

---

### Official Review · Reviewer_dePm · 2021-07-16

**Rating:** 7
**Confidence:** 3

**Summary:**

The paper gives a variety of backdoor-related theoretical results. In particular it,
1. gives a theoretical framework for discussing backdoor poisoning attacks,
2. shows that the attacker does not need to know the true data distribution to launch an attack against an ERM learner,
3. introduces _memorization capacity_, which measures the amount of off-distribution data a learner can memorize, and shows that nonzero memorization capacity is both necessary and sufficient for the existence of a backdoor attack
4. (the attack constructed in 4. can succeed with limited poisoned data regardless of how much clean data there is),
5. under assumption of finite VC-dimension, shows that adversarial training can be used to detect backdoor attacks, and
6. under the same assumption, shows that the problem of learning a backdoor-resistant model is equivalent to the problem of filtering poisoned data from the dataset.


**Limitations And Societal Impact:**

They have.

**Main Review:**

The paper multiple noteworthy theoretical results relating to backdoor poisoning attacks.

Typos:
287: Assumption on VC-dimension is incomplete.


**Time Spent Reviewing:**

2

---

> ### Author Response · Authors · 2021-08-10
> **Response to Official Review of Paper4597 by Reviewer dePm**
>
> Thank you for the positive review and for catching the typo on line 287. Line 287 should have read as "the assumption that $\mathsf{VC}\left(\mathcal{L}_{\mathcal{F}\mathsf{adv}(h^{*})}\right) < \infty$." We will fix this for the camera-ready version.

---

### Official Review · Reviewer_jukN · 2021-07-16

**Rating:** 8
**Confidence:** 4

**Summary:**

This work proposes a theoretical framework to draw connections between several phenomena in machine learning observed empirically but never adequately understood- the relationship between excess model capacity, perturbation robustness, and backdoor poisoning. They show how under specific settings, adversarial training can detect backdoored data or recover a classifier robust to the backdoors at the very least. Similarly, they show how learning a backdoor-robust classifier is equivalent to pruning corrupted train data. They further demonstrate the applicability of their theorems on the MNIST dataset.

**Limitations And Societal Impact:**

The authors could certainly extend their empirical evaluation to more complex datasets in the near future. Still, I do not see any obvious limitations in its current state except the ones already stated in their assumptions. Nonetheless, it would be interesting how this could be utilized to perform "adversarial training" indirectly - especially for text-related tasks.

**Main Review:**

## Positive Feedback

1. This work has instrumental theoretical contributions: in terms of findings, theorems, the framework itself, and equivalence between robustness and susceptibility to poisoning.

2. Problem 2  has been formulated well, with an easy-to-follow breakdown of the adversary's goals. Having equivalent text explanations next to formulae is very helpful.

3. The fact that results from the proposed framework align with empirical observations in this field by other authors reinforce this work's soundness and helps better understand the reason behind those phenomena.

---

## Criticism

1. Empirical results help show that the proposed theory extends to implementation (which can sometimes lead to significant drops because of stochasticity in model training and constants in equations). Given that the authors did perform such experiments (Supplementary material), it would be great to have a small sub-section summarizing those empirical results.

2. Line 208: "..a 0-loss classifier" . Would this result hold if the focus were to find some $\epsilon$-loss classifier for some reasonably small $\epsilon$? I am a bit concerned about this part since, in a realistic training scenario, the model training would rarely find a 0-loss set of parameters (and even if it could, would never want 0 loss on training data).

---

## Minor comments

1. Line 1: "...supervised learning" is not a requirement for backdoor data poisoning - the setting can also be unsupervised or semi-supervised. Lines 1-7: Do not need such an extended portion (that too in the abstract) dedicated to this definition. In general, the Abstract could use a rewrite - it is too long right now and captures unnecessary details.

2. Lines 26-28: can skip, is pretty common knowledge in this field by now.

3. Line 3: "...(nearly) 0 error" not sure I agree with this - wouldn't having a training error too low be akin to overfitting, which is something machine-learning algorithms actively avoid.

4. Section 1.2: It is nice to see that the authors include this, but it would probably be best to shift it to later in the paper.

5. Theorem 17 uses $\alpha\leq\frac{1}{3}$ and Theorem 18 uses $\alpha\leq\frac{1}{6}$. My understanding is that the $2\alpha$ fraction from Theorem 18 connects to Theorem 17? If yes, it might be worthwhile to write them accordingly before combing the two Equations to imply equivalence because, at the moment, both Equations have different constraints.


## Update

Increased confidence to 4

**Time Spent Reviewing:**

4

---

> ### Author Response · Authors · 2021-08-10
> **Response to Official Review of Paper4597 by Reviewer jukN**
>
> Thank you for your positive and detailed review. Please see our responses to your individual suggestions below.
>
> _"Given that the authors did perform such experiments (Supplementary material), it would be great to have a small sub-section summarizing those empirical results."_
>
> We agree with this, and we'll be happy to write a small sub-section immediately following lines 298-301 summarizing our empirical results.
>
> _"Line 208: "..a 0-loss classifier" . Would this result hold if the focus were to find some -loss classifier for some reasonably small $\varepsilon$? I am a bit concerned about this part since, in a realistic training scenario, the model training would rarely find a 0-loss set of parameters (and even if it could, would never want 0 loss on training data)."_
>
> First, we address the concern regarding a $0$ training loss being undesirable. In our scenario and under our assumptions, finding a classifier within $\mathcal{H}$ minimizing error on the training set is actually equivalent to obtaining a classifier that generalizes well to unseen data. This is because we assume that $\mathsf{VC}(\mathcal{H}) < \infty$, which then implies that with sufficiently many samples (roughly $O(\mathsf{VC}(\mathcal{H}))$), the performance of every $h \in \mathcal{H}$ on the training set is close to the true performance of $h$, including for the $h$ that minimizes the training set error. In other words, with enough training samples (this quantity is explicitly stated in the relevant theorem statements), we achieve the uniform convergence property.
>
> With this in mind, we can now discuss the condition on Line 208. Our assumption here is that there exists a $0$-loss classifier on the data distribution (and therefore there exists a perfect classifier on the training set). Per the previous argument, as long as we achieve the uniform convergence property, this is actually OK -- in particular, with high probability, the training loss will be pretty close (up to $\varepsilon_{\mathsf{clean}}$ or $\varepsilon_{\mathsf{adv}}$) to the test loss. With that said, as you suggest, extending these results to the agnostic case (that is, the case where there might not be a perfect classifier on the data distribution) is a very interesting direction for future work.
>
> _"Line 1: "...supervised learning" is not a requirement for backdoor data poisoning - the setting can also be unsupervised or semi-supervised. Lines 1-7: Do not need such an extended portion (that too in the abstract) dedicated to this definition. In general, the Abstract could use a rewrite - it is too long right now and captures unnecessary details._"
>
> Thank you for the comment. It's a good point that the supervised vs unsupervised vs semi-supervised distinction doesn't actually matter in the grand scheme of things (though it does for our exposition), and we'll fix this. More generally, we'll try to shorten the abstract as possible.
>
> _"Lines 26-28: can skip, is pretty common knowledge in this field by now._"
>
> Thanks for letting us know. We'll take care of this for the camera-ready version.
>
> _"Line 3: "...(nearly) 0 error" not sure I agree with this - wouldn't having a training error too low be akin to overfitting, which is something machine-learning algorithms actively avoid."_
>
> Did you mean Line 37? As for the specific claim -- although there are settings in which $0$ error may indeed be a sign of overfitting, it has been observed empirically in neural network learning problems where the learned classifier generalizes well to the test set despite perfectly fitting the training set. This also holds true when there is a backdoor attack being executed (e.g. see [13], [16]). In any case, we'll be happy to change this phrasing to say "... low error."
>
> _"Section 1.2: It is nice to see that the authors include this, but it would probably be best to shift it to later in the paper."_
>
> Sure, will do.
>
> _"Theorem 17 uses $\alpha \le \frac{1}{3}$ and Theorem 18 uses $\alpha \le \frac{1}{6}$. My understanding is that the $2\alpha$ fraction from Theorem 18 connects to Theorem 17? If yes, it might be worthwhile to write them accordingly before combing the two Equations to imply equivalence because, at the moment, both Equations have different constraints."_
>
> Thank you for raising this concern. To clarify, a loose restatement of Theorem 18 can go something like "a generalization algorithm with $2\alpha$-outlier tolerance can be converted to a filtering algorithm with $\alpha$-outlier tolerance." As an artifact of our current analysis, we use the factor-$2$ separation between the two $\alpha$ values in Theorems 17 and 18. We will clarify this limitation in the camera-ready version of the paper.
>
> _Limitations and Societal Impact_
>
> Thank you for the suggestions here. Indeed, a more comprehensive empirical evaluation of Theorem 14 is an interesting future direction, both on other image datasets and on speech/language tasks.

---

> > ### Comment · Reviewer_jukN · 2021-08-23
> > **Acknowledgement**
> >
> > Thanks to the authors for their response. I had very few issues with the paper, which I feel have been addressed. This is a good paper, and should definitely be part of the conference's proceedings :) I have updated my rating to reflect the same.

---

### Official Review · Reviewer_bwBK · 2021-07-19

**Rating:** 6
**Confidence:** 3

**Summary:**

The authors propose a theoretical study on the robustness of binary classifiers against backdoors. To this end, they provide a theoretical framework and define a measure to assess classifiers' memorization capacity. This formulation allows the authors to demonstrate the existence of backdoors in some settings and explain why adversarial training reduces the vulnerability to backdoors.

**Ethics Review Area:**

["I don’t know"]

**Limitations And Societal Impact:**

Potentially-negative societal impact and limitations have been discussed.

**Main Review:**

The main contribution of this work is the framework devised by the authors to explain some aspects underlying the success or failure of backdoor attacks. While related work in the area of backdoor attacks is well acknowledged and discussed, the authors should better position this work also with respect to other theoretical analyses related to the memorization of training samples and to the relationship between over-parameterized models and their ability to fit random labels. In particular, from this manuscript, the memorization of the training points seems a concept discovered and defined by the authors. However, this is not the case, as previous work has defined this concept [a] and provided a way to estimate training point memorization [b]. The authors should also clarify that it is already known that robust training increases the robustness against backdoors [c].
Last but not least, it would be really a plus if the authors could somewhat connect their work to these papers:
- https://arxiv.org/abs/1611.03530,
- https://arxiv.org/abs/2105.14368 (Sect. 5.2 and reference [10]).

Notwithstanding, this manuscript has some novelty. To my knowledge, the concept of memorization capacity has never been related before to the vulnerability against backdoors, and the validity of adversarial training as a defense against this attack was empirically shown but not theoretically explained. The formulation devised by the authors may be adapted in the future to understand better other phenomena related to machine learning robustness against attacks.

I appreciated the authors' effort to explain the proposed formulation clearly. To be a theoretical paper is pleasant to read and easy to understand, even though a bit dense in some parts. I would also recommend the authors to try to better emphasize and recap (e.g., in the conclusions) constructive ways in which their formulation can be adopted to improve current algorithms/best practices.

Overall, I think this work provides a fair contribution to the state of the art.

Minor comments:

As a minor comment, the first work introducing backdoor attacks is not [6] but:
- T. Gu, B. Dolan-Gavitt, and S. Garg. Badnets: Identifying vulnerabilities in the machine learning model supply chain. In NIPS Workshop on Machine Learning and Computer Security, abs/1708.06733, 2017.


References

a) Arpit, Devansh, Stanisław Jastrzębski, Nicolas Ballas, David Krueger, Emmanuel Bengio, Maxinder S. Kanwal, Tegan Maharaj, et al. ‘A Closer Look at Memorization in Deep Networks’. In Proceedings of the 34th International Conference on Machine Learning - Volume 70, 233–42. ICML'177. Sydney, NSW, Australia: JMLR.org, 2017.

b) Feldman, Vitaly, and Chiyuan Zhang." What Neural Networks Memorize and Why: Discovering the Long Tail via Influence Estimation". In Advances in Neural Information Processing Systems, edited by H. Larochelle, M. Ranzato, R. Hadsell, M. F. Balcan, and H. Lin, 33:2881–91. Curran Associates, Inc., 2020. https://proceedings.neurips.cc/paper/2020/file/1e14bfe2714193e7af5abc64ecbd6b46-Paper.pdf.

c) Borgnia, Eitan, Valeriia Cherepanova, Liam Fowl, Amin Ghiasi, Jonas Geiping, Micah Goldblum, Tom Goldstein, and Arjun Gupta. 'Strong Data Augmentation Sanitizes Poisoning and Backdoor Attacks Without an Accuracy Tradeoff'. ArXiv:2011.09527 [Cs], 18 November 2020. http://arxiv.org/abs/2011.09527.


**Time Spent Reviewing:**

4h

---

> ### Author Response · Authors · 2021-08-10
> **Response to Official Review of Paper4597 by Reviewer bwBK**
>
> Thank you for your positive and detailed review. Please see our responses to your individual suggestions below.
>
> _"The authors should better position this work also with respect to other theoretical analyses related to the memorization of training samples and to the relationship between over-parameterized models and their ability to fit random labels."_
>
> Thank you for the suggestion and the pointers to related literature. We will include a greater discussion in the related works section connecting our work to prior discussions of memorizing samples and fitting random labels.
>
> _"In particular, from this manuscript, the memorization of the training points seems a concept discovered and defined by the authors."_
>
> We do not claim to be the first to discover the ability of ML classifiers to memorize training data points. However, as far as we are aware and as you point out, our work is the first to relate off-distribution memorization to the vulnerability of a learning problem to backdoor attacks. We certainly didn't intend to suggest otherwise and will make sure this is clear in the final version.
>
> _"The authors should also clarify that it is already known that robust training increases the robustness against backdoors [c]."_
>
> Thank you for bringing this paper to our attention; we will mention this empirical contribution prior to our statement of Theorem 14. As you point out, [c] does show that a form of robust training can increase the robustness against backdoors at test-time. However, Theorem 14 of our work yields a converse as well -- in particular, under the assumption that there exists a perfectly robust classifier on the data distribution, if the robust loss at train-time is high, then the learner can be pretty certain that there are malicious corruptions present in the training set. This can prompt human inspection of the dataset or the usage of some data sanitization algorithm as applicable to the specific scenario at hand. To our knowledge, our work is the first to establish this converse.
>
> _"Last but not least, it would be really a plus if the authors could somewhat connect their work to these papers"_
>
> Thank you for the pointers; we will discuss these works in our related works section. As an aside, it is intuitive that there are interesting connections between our work, the ability to memorize randomly labeled examples, and test-time adversarial examples. However, we are unsure of what nontrivial, precise statements can be made to these effects. Fleshing out these relationships constitutes a very interesting direction for future work.
>
> _"I would also recommend the authors to try to better emphasize and recap (e.g., in the conclusions) constructive
> ways in which their formulation can be adopted to improve current algorithms/best practices."_
>
> We'll be happy to include a discussion to this effect in the conclusion.
>
> _"As a minor comment, the first work introducing backdoor attacks is not [6] but"_
>
> Thank you for catching this; we will update this statement in the camera-ready version of our paper.

---

> > ### Comment · Reviewer_bwBK · 2021-08-30
> > **Acknowledgement**
> >
> > I would like to thank the authors for their responses, and I am looking forward to reading their updated work.

---

### Decision · Program_Chairs · 2021-09-27

**Decision:**

Accept (Spotlight)

**Comment:**

All reviewers agree that this is an important result filling in the gap of theoretically understanding backdoor attacks.